# HyLight: Strain aware assembly of low coverage metagenomes

Xiongbin Kang [1,2], Wenhai Zhang [1], Yichen Li[3], Xiao Luo [1]✉ &
Alexander Schönhuth [2]✉

Different strains of identical species can vary substantially in terms of their
spectrum of biomedically relevant phenotypes. Reconstructing the genomes
of microbial communities at the level of their strains poses significant chal-
lenges, because sequencing errors can obscure strain-specific variants. Next-
generation sequencing (NGS) reads are too short to resolve complex genomic
regions. Third-generation sequencing (TGS) reads, although longer, are prone
to higher error rates or substantially more expensive. Limiting TGS coverage to
reduce costs compromises the accuracy of the assemblies. This explains why
prior approaches agree on losses in strain awareness, accuracy, tendentially
excessive costs, or combinations thereof. We introduce HyLight, a metagen-
ome assembly approach that addresses these challenges by implementing the
complementary strengths of TGS and NGS data. HyLight employs strain-
resolved overlap graphs (OG) to accurately reconstruct individual strains
within microbial communities. Our experiments demonstrate that HyLight
produces strain-aware and contiguous assemblies at minimal error content,
while significantly reducing costs because utilizing low-coverage TGS data.
HyLight achieves an average improvement of 19.05% in preserving strain
identity and demonstrates near-complete strain awareness across diverse
datasets. In summary, HyLight offers considerable advances in metagenome
assembly, insofar as it delivers significantly enhanced strain awareness, con-
tiguity, and accuracy without the typical compromises observed in existing
approaches.

Metagenomics has revolutionized our understanding of microbial
diversity and functional potential in various environments by
sequencing the collective genomic material of microbial communities
in many niches. The advent of high-throughput sequencing technol-
ogies (NGS) has made metagenomic research increasingly accessible,
providing valuable insight into complex microbial ecosystems[1–3].

The assembly of metagenomic data poses significant challenges
due to the presence of multiple strains, high levels of genetic
diversity, and varying abundances of different organisms within a
community[4,5]. Strains that appear at only low abundance in a

metagenome, but whose discovery is crucial in the context of under-
standing the environment under investigation, stress the practical
relevance of these challenges.

Importantly, strains can display significant variations in their
interactions and environmental impact[6,7]. Strains can also differ in
clinically relevant aspects such as medication resistance, virulence,
and host-microbiome interactions[8–11]. For these reasons, it is crucial to
identify genomes at strain resolution. Therefore strain aware meta-
genome assembly has become the driving technical and methodical
challenge[12,13].

---

[1]College of Biology, Hunan University, Changsha, China. [2]Genome Data Science, Faculty of Technology, Bielefeld University, Bielefeld, Germany. [3]College of
Computer Science and Electronic Engineering, Hunan University, Changsha, China. ✉e-mail: xluo@hnu.edu.cn; aschoen@cebitec.uni-bielefeld.de

Next-generation sequencing (NGS), where Illumina sequencing technology is the leading option, is extensively utilized in metagenomics research for being culture-independent, and its combination of high throughput, low error rates (below 1%), and cost-effectiveness[14]. Short reads typically ranging from only 35 bp to 700 bp in length[15] induces obvious difficulties in spanning complex genomic regions[16,17], which leads to fragmented genome assemblies[18–20]. The limitation becomes particularly concerning when processing metagenomic samples of greater complexity, characterized by harboring diverse strains[19]. In summary, the reconstruction of both strain-level resolved and complete genomes of individual organisms from metagenomes is too challenging when using NGS alone.

Third-generation sequencing (TGS), such as Pacific Biosciences (PacBio) and Oxford Nanopore Technologies (ONT), holds the promise to overcome these challenges. TGS reads are significantly longer, so they can also span complex regions, which makes it possible to distinguish between related organisms[13,15,21,22]. The disadvantages of TGS are the considerably higher error rates and substantially larger costs. Even the relatively cheap PacBio CLR and ONT reads imply substantially larger costs because their high error rates (up to 15%) require elevated coverage to compensate the errors[21].

Latest generations of PacBio (HiFi) or ONT (Q30+) reads are both significantly longer than NGS reads and contain similarly little errors. However, they do not alleviate all relevant issues. In comparison with noisy PacBio CLR or ONT their length is substantially shorter (often exceeding 10 kb, but being up to 4-8 times shorter than PacBio CLR or ONT reads[21]). Also, they come at lower throughput in general while still drawing substantially larger costs[21]. The computational bottlenecks explain why hardware-based solutions, for example GPU-tailored versions of the algorithms supporting PacBio HiFi and most recent ONT reads algorithm are now considered. Of course, this requires various sequencing laboratories to re-consider their hardware equipment, which leads to further investments. In summary, PacBio HiFi or recent generations of ONT type reads mean practical convenience only sufficiently generously budgeted laboratories can afford. The vast majority of sequencing laboratories, however, keeps depending on NGS on one hand, and cheaper hence more erroneous (but in case of PacBio CLR and ONT still the longest) types of TGS reads, and running their evaluation of sequencing experiments on standard computational machinery. In summary, sole application of TGS in metagenomics requires to substantially raise expenses: either to increase coverage (as for PacBio CLR and ONT) or because of employing more sophisticated sequencing protocols (PacBio HiFi or ONT reads of latest generations)[21,23,24]. The great majority of labs worldwide cannot employ TGS (regardless of the particular type) alone for strain-level metagenome assembly: either coverage remains too low or expenses become excessive. This hampers the routine and widespread implementation of successful strain-aware metagenome analyzes in particular in disease research.

The mix of theoretical and practical challenges just outlined is the explanation for why StrainXpress[12] which is based on NGS alone, and Strainberry[13] which is based on only TGS, are the only two approaches that decidedly focus on de novo strain aware metagenome assembly. Both StrainXpress and Strainberry provide important inspiration, as they both employ techniques that had hitherto never been considered for metagenome assembly. While StrainXpress employs overlap graphs, Strainberry is based on the minimum error correction problem to separate strain-specific haplotypes in an iterative scheme (see Background below for details). Still, neither of the two approaches generates assemblies of the kind of quality that we are envisioning. While StrainXpress assemblies remain too fragmented, Strainberry requires elevated coverage rates for the TGS reads (which, as repeatedly pointed out, is expensive). This confirms that application of only NGS or application of only TGS is insufficient from also a methodological point of view. We recall, as repeatedly pointed out, that resorting to standard techniques runs into larger costs or lower assembly quality, or even both of them.

Here, our goal is to provide a de novo strain-aware metagenome assembly approach that does not only push the limits in terms of assembly quality, but is also as inexpensive as possible. The latter point ensures that the majority of laboratories worldwide can afford related sequencing strategies. The first point potentially puts our approach also in the focus of amply budgeted laboratories, if they aim at optimal assembly quality. To reach this goal, we are guided by the insight that, recently, methodical novelties were the only way to make progress.

The basis for our strategy is hybrid assembly. Hybrid assembly synthesizes the advantages of short and long reads. Short and long reads complement each other perfectly, insofar as their advantages mutually cancel their disadvantages. While NGS reads are accurate and short, TGS reads are inaccurate and long. Therefore, synthesizing short and long reads yields long and accurate fragments, which is the optimal basis for high-quality assemblies. The practical feasibility of hybrid assembly is established by the fact that the vast majority of sequencing laboratories are equipped with (Illumina type) NGS platforms and earlier-type TGS platforms. Both such platforms deliver reads of coverage sufficient for our approach at small expenses.

The accuracy of the assemblies and the inexpensiveness of the supporting data have already been noticed in prior work. Despite their benefits, all current state-of-the-art hybrid metagenome assembly approaches[25–28] operate at the species, but not at the strain level, as their finest taxonomic resolution. All these approaches have been widely applied in studies that require metagenomic assembly as an essential step[29–33]. This documents the popularity of hybrid assembly approaches already when being used for only identifying species, but not yet strains.

Understanding why the state-of-the-art of hybrid approaches only delivers species-resolved metagenome assemblies requires a look at the underlying methodologies. One realizes that both specialized hybrid metagenome as well as more generic hybrid assemblers broadly fall into two categories: short-read-first and long-read-first approaches. Short-read-first approaches assemble short reads first, and then scaffold the short read based contigs guided by the long reads. Vice versa, long-read-first approaches assemble only the long reads, and treat short reads as auxiliary source of information. Thereby, they align short reads either against the already assembled long reads to eliminate the errors from the contigs[34], or against the raw, unassembled long reads to eliminate the errors prior to their assembly[35]. Importantly, none of the approaches so far presented assembles both long reads guided by short reads and short reads guided by long reads, so as to synthesize the advantages of the two axes of approaches. The likely reason for this is the potential complexity in terms of protocols that such "cross-hybrid" or "mutual support" strategies may entail.

There are two additional important observations. First, all long-read-first assemblers so far presented do not specialize in the assembly of metagenomes. This means that all current state-of-the-art in hybrid metagenome assembly[25–28] employ short-read-first type strategies. Second, all short-read-first approaches are based on de Bruijn graphs (DBG's) as the underlying assembly paradigm. The reason for the latter is the fact that DBG's are by far the predominant assembly paradigm when processing short reads. See Background below for more details.

In an encompassing summary of the state-of-the-art, one concludes that (1) there are no hybrid metagenome assemblers that address strain awareness, (2) there are no hybrid metagenome assemblers that build on overlap graphs as their assembly paradigm, (3) there are no long-read-first approaches to hybrid metagenome assembly, and (4) there are no hybrid assemblers that build on both short-read- and long-read-first type strategies in combination. This explains why we focus on the non-hybrid, but strain aware metagenome assemblers StrainXpress[12] and Strainberry[13] and on the hybrid, but non-strain-aware metagenome assemblers Opera-MS[27], HybridSPAdes[25], MetaPlatanus[28] and Unicycler[26] in our benchmark experiments.

Based on these observations, we suggest a strategy that neither reflects a short-read-first nor long-read-first approach. Instead, we suggest to compute assemblies from both the long reads and the short reads. Thereby, short reads assist in the assembly of the long reads and vice versa. So, we treat both long and short reads as both primary assembly and auxiliary data. Upon having assembled both long and short reads, we merge the two assemblies into a unifying set of scaffolded contigs. As a key point, we avoid excessively complex assembly protocols by not assembling batches of reads whose assemblies can be foreseen to be redundant. To the best of our understanding, enabling the usage of both short and long reads as primary (assembly) and secondary (auxiliary) data via a sufficiently lightweight protocol establishes a methodical novelty. From this perspective, our approach adds a third category, which one could refer to as "cross hybrid" or, alternatively, "mutual support" approach to the well-established categories of short-read-first and long-read-first approaches. It may be a relevant observation that our approach is also an advance insofar as it is a hybrid metagenome assembly approach that makes use of overlap graphs, and not DBG's, as the unifying data frame. See again Background just below for full details.

## Background

### Strain aware assemblers: strainxpress, overlap graphs and Strainberry

*StrainXpress* employs overlap graphs (OG's) instead of de Bruijn graphs (DBGs) where the latter ones have been the (by far) predominant data structure paradigm employed for NGS based assembly. The crucial insight here is the fact that usage of OG's effectively aids in spanning complex repetitive regions, because OG's do not require to chop reads into $k$-mers. As a consequence, genetic linkage of strain specific variants becomes evident. This makes it possible to extend contigs across regions that remain "strain-specific variant deserts" when operating with NGS reads or de Bruijn graphs.

*Note on Overlap Graphs:* Our approach draws inspiration from StrainXpress, which demonstrated that OG's could also be used in an advantageous way when it comes to distinguishing between similar strains of the same species. To understand this better, we recall that the construction of DBG's implies to chop reads into k-mers. The artificial shortening of the reads leads to significant losses in terms of information with respect to genetic linkage of strain specific variants (in particular, one can no longer trace linkage of variants at distance larger than $k$ when employing $k$-mers[36]). Unlike DBG's, OG's preserve the haplotype (strain) identity of the reads to a maximum degree, which we systematically exploit also here. On a side remark, note that the employment of OG's also lead to superior strain awareness in viral quasispecies assembly[37,38].

*Strainberry*, on the other hand, initially employs Metaflye[39] to assemble the TGS reads into contigs, and subsequently aligns the long reads with the contigs. Based on these alignments, Strainberry calls SNP's where contigs serve to provide auxiliary reference coordinates. The resulting scenario provides the basis for solving the minimum error correction problem to phase reads into haplotypes. The source of inspiration is earlier work that proved that modeling haplotype separation as instances of the minimum error correction problem was useful[40–42]. As a methodical novelty, Strainberry applies this procedure in the frame of an iterative protocol: the iteration ends when separation does no longer reveal new haplotypes; eventually strain-resolved contigs are assembled using Wtdbg2[43].

### Hybrid metagenome assemblers: workflows

In a bit more detail, Opera-MS[27] first uses an established short-read metagenome assembler like MegaHit[44], MetaSPAdes[45] or IBDA-UD (all of which are DBG based[46]) to assemble the short reads. Subsequently, both short and long reads are mapped against the short-read contigs to obtain coverage and linkage information for the contigs, and to construct an assembly graph based on that information. Contigs are further hierarchically clustered using a distance measure that reflects the distance between the contigs in terms of their distance in the assembly graph. Apart from certain details, each cluster is supposed to collect the contigs of one species. Finally, contigs within clusters are scaffolded using Opera-LG, which was designed to work for the hybrid assembly of isolated genomes.

HybridSpades[25] constructs an assembly graph from the short reads using SPAdes[47], by removing bulges, tips and chimeric edges. By mapping the long reads against the resulting assembly graph, it generates read paths, and further closes gaps in the assembly graph by using the consensus of the long reads that span the gaps. Then, by extending a technique that addresses growing read paths by trying "extension edges" to incorporate long read paths, it resolves repeats.

MetaPlatanus[28] computes contigs by constructing a DBG, and subsequently corrects contigs based on coverage considerations relating to both short and long reads, and also untangles "cross-structures" from such considerations. DBG's are re-constructed iteratively, where in each iteration the improved contigs serve as the basis for constructing the DBG, and subsequent correction of contigs yields new contigs. MetaPlatanus then scaffolds the resulting contigs using long read links, and, possibly, re-constructs the DBG another last time. Eventually, scaffolds are binned, where each bin is supposed to refer to a particular species. Within bins, gaps are closed, edges are extended, and possibly unused short reads are employed for aiding in that. In the ultimate step, gaps are closed, and scaffolds are polished using techniques that address the generation of assemblies for isolate genomes.

Unicycler[26], like HybridSPAdes, uses SPAdes[47] for assembling the short NGS reads. Like MetaPlatanus, it employs coverage considerations to refine the resulting assembly graph. Subsequent integration of long reads then points out paths in the refined assembly graph. "Bridges" that reflect resulting new links in the assembly graph are ranked by quality (measured in terms of coverage, for example), and then are applied in decreasing order relative to their quality, which establishes the paths that are supposed to be real. The resulting assemblies are finally polished by re-aligning the short reads against the selected paths using Bowtie[48], as a standard short-read mapper.

### Summary of contributions

In summary, the advances we suggest are as follows.

1. Here we show the, to the best of our knowledge, first hybrid metagenome assembly approach that is strain aware.

2. In all earlier approaches, either short reads or long reads were used as fundamental assembly data where the other type of reads was used as auxiliary data. We suggest a hybrid assembly approach that makes use of both long and short reads as both primary (assembly) and secondary (auxiliary) data. In other words, one can arguably refer to our approach as being "cross-hybrid" in nature.

3. We suggest the, to the best of our knowledge, first hybrid assembly approach in which overlap graphs are used to capture effects relevant for the assembly of the short reads. A particular feature is the employment of "contig OGs", as a rather unusual concept in genome assembly.

4. We suggest a metagenome asssembly approach that is strain aware without requirements in terms of long read coverage or sophisticated sequencing protocols that tend to be excessive in terms of costs.

5. Last but not least, we suggest a metagenome assembly approach that is superior over all prior approaches in terms of assembly quality. Arguably, by its results, HyLight considerably pushes the limits of possibilities in strain aware metagenome assembly, all in terms of strain awareness, contig length (contiguity), and in terms of the accuracy of the contigs.

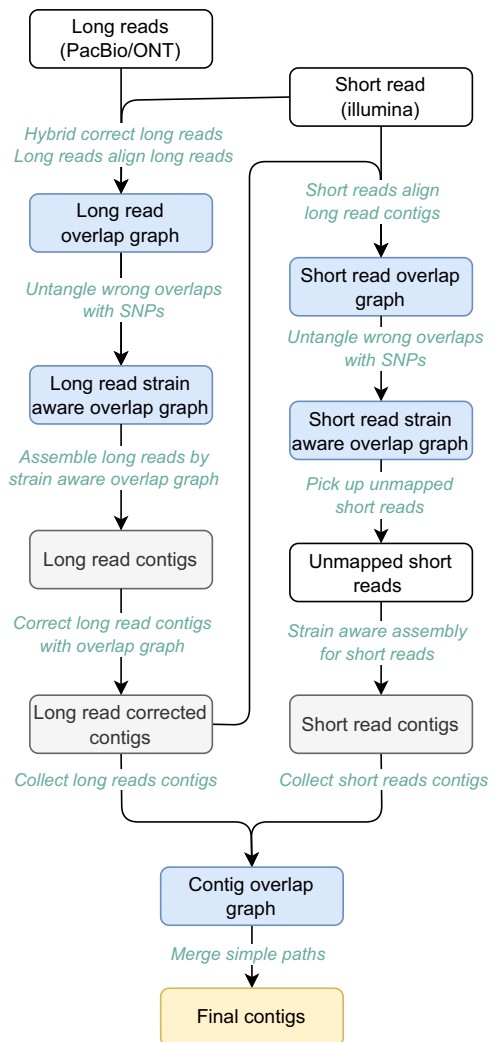

**Fig. 1 | Workflow of HyLight.** The input data consists of two fastq files, long reads and short reads. The output is a fasta file containing the assembled contigs. The overall procedure can be divided into three primary steps. Firstly, strain-resolved OG is conducted to assemble long reads. Subsequently, another OG is established to assemble short reads. Lastly, a contig OG is established to extend the contigs obtained from the assembly results of long and short reads, culminating in the generation of the final master contigs.

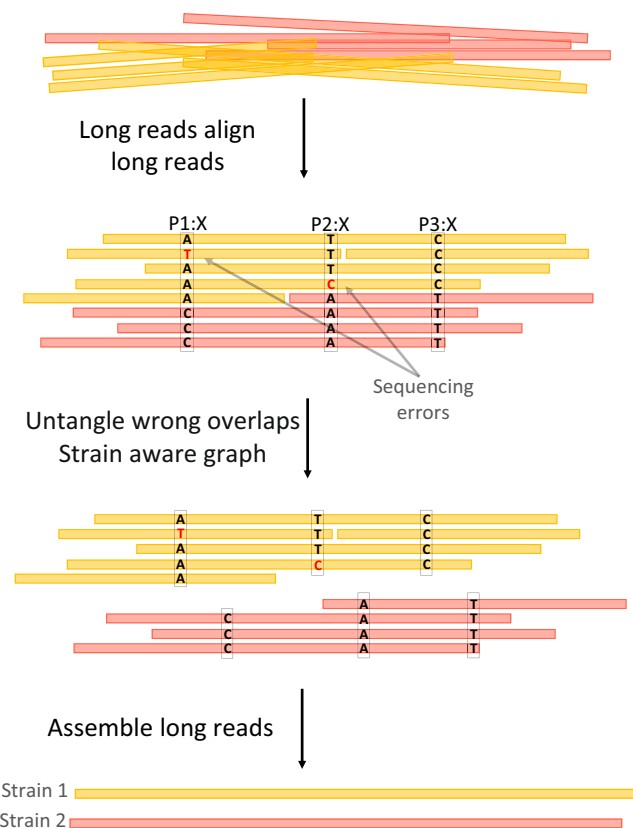

**Fig. 2 | Assembly of long reads.** Long reads are pre-corrected using short reads. The distinct colors of the reads indicate their respective strain origins. The objective of this workflow is to leverage SNP information to filter out incorrect overlaps, selectively retaining overlaps between reads originating from the same strain. This enables strain aware assembly to be performed effectively.

In the following, in Results, we will present the workflow of our approach as well as all details required for understanding it from a larger perspective. Subsequently, we will present the experiments that confirm the novelties and improvements as pointed out above. As usual, we will discuss our results in comparison with the hypotheses just raised. In Online Methods, we will provide all the details that establish the full reproducibility of our approach.

## Results

In the following, we first briefly discuss the workflow. Details that are necessary to fully reproduce the workflow from a conceptual point of view are provided in Methods. Subsequently, we present experiments on both simulated and real data, which provide evidence for the benefits of our approach, as listed towards the end of the Introduction.

HyLight's major innovation lies in the construction of a strain-resolved overlap graph (OG) as input for assembling long reads, correcting contigs, and clustering and assembling short reads, ultimately achieving strain-aware assembly.

### Workflow

Please see Fig. 1 for a schematic of the workflow. The workflow proceeds in two axes: one for assembling the long reads (see left branch in Fig. 1) and one for assembling the short reads (right branch in Fig. 1). Assemblies of long and short reads are merged in a final step (see bottom of Fig. 1).

In a brief summary, HyLight performs the following items. First, it corrects long reads using short reads, which turns the raw TGS reads into polished, error-free long reads. The resulting polished long reads then are the basis for constructing a strain-resolved overlap graph that gets further polished by removing remaining errors.

To provide an overview of the workflow, this section offers a high-level description. For detailed descriptions of all methodical steps involved, please refer to the "Methods" section. Figure 1 illustrates the overall workflow of HyLight. As outlined previously, HyLight comprises three main modules.

**First module: long read assembly.** The main purpose of this axis is to compute strain-aware, error-free contigs from the long reads.

1. Long reads are corrected using short reads using FM-index and de Bruijn graph based techniques, as implemented in FMLRC2[49], which has been shown to outperform other methods in recent benchmark studies[50,51].
2. An overlap graph is constructed from the corrected long reads. We make use of the (widely popular) Minimap2[52] to compute the necessary overlaps.
3. We identify overlaps that connect long reads from different strains by inspecting SNP patterns, and we remove edges in the overlap graph that reflect the connection of long reads from

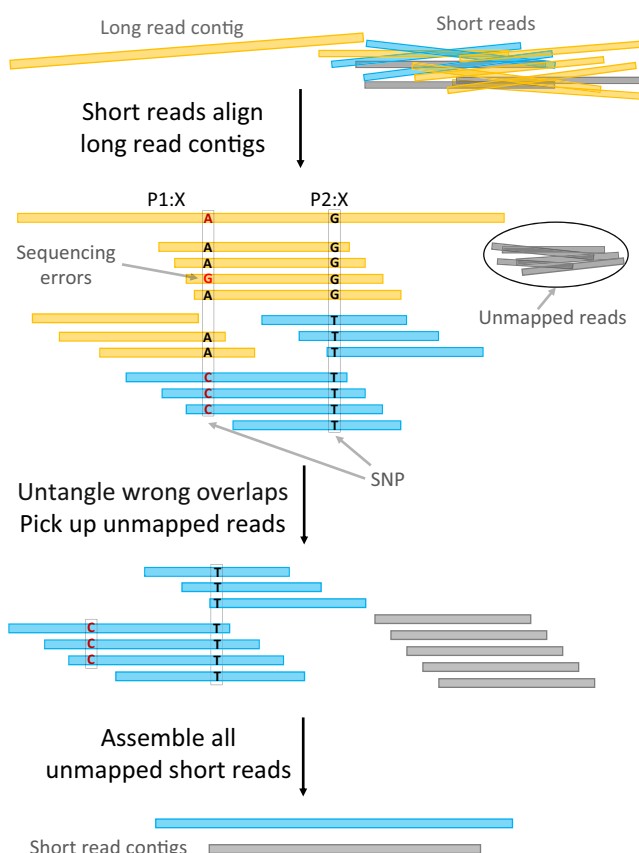

**Fig. 3 | Assembly of short reads.** The distinct colors of the reads indicate their respective strain origins. The primary procedure consists of aligning the short reads to the contigs, establishing a strain-resolved OG, and then excluding short reads that align to regions already assembled into contigs. Subsequently, an OG is constructed to assemble the remaining short reads and reconstruct strains or regions that were not initially assembled.

different strains. See Fig. 2 for an illustration. The result is an overlap graph of the long reads that consists of connected components each of which contains long reads from only one particular strain. So, each connected component in the overlap graph now reflects a collection of reads drawn from one haploid genome.

4. We assemble the long reads based on the resulting strain-aware overlap graph using Miniasm[53], which is a long read assembler that addresses to assemble haploid genomes from long reads. The result are contigs each of which stems from one particular strain.
5. We re-align the long reads against the resulting strain-aware contigs.
6. Based on the re-alignment, we establish a second, improved version of an overlap graph for the long reads, which now reflects a strain-aware overlap graph of the long reads.
7. Using this improved, strain-aware overlap graph, we remove errors that have remained in the long reads using Racon[54].

As for 7, note that Racon has not been designed to operate in a strain-aware manner. If one fed the original, strain-unaware overlap graph to Racon, it would "overcorrect" contigs by mistaking true, strain-specific variation for errors and eliminating them. This would mask strain-specific variation hence prevent the reconstruction of genuine strain-specific sequence. One can consider the application of Racon to only strain-aware overlap graphs an insight that is crucial for computing both error-free and strain-aware long read based assemblies.

**Second module: short read assembly.** We recall that it is a general objective to establish a workflow that caters to low coverage of long reads to avoid unnecessary and possibly unaffordable costs. Therefore, the main purpose of this axis is to assemble the (likely high coverage, because cheap) short reads in their own right, and use the assemblies to fill gaps in the long read based assembly, or even identify additional strains from the resulting contigs.

1. We align the short reads against the strain-aware, error-free contigs, as the output of the long read axis (first module), using Miniasm[53].
2. The alignment of short reads with long read contigs gives rise to an overlap graph of the short reads.
3. Analogously to the long read axis, we inspect SNP patterns in the overlap of the short reads. Based on the SNP patterns, we identify overlaps of short reads that reflect to connect two short reads from different strains. See Fig. 3 for an illustration.
4. As a result, we are now able to identify short reads whose SNP patterns contradict their initial alignment with the long read contigs. The insight is that breaking up overlaps between short reads all of which align with the same long read contig leads to several classes of short reads. Only one of the classes of short reads truly agrees with their respective long read contig (yellow short reads in Fig. 3).
5. We conclude that short reads no longer having overlaps with short reads whose SNP patterns truly match those of their long read contigs, do not stem from the same strain as the long read contig against which they initially aligned (blue short reads in Fig. 3).
6. Further, we collect all short reads that did not align with any of the long read contigs (gray short reads in Fig. 3).
7. We discard all short reads whose alignments indicated full agreement with a long read contig (yellow in Fig. 3).
8. Subsequently, using StrainXpress[12] (which as discussed in the Introduction specializes in the strain-aware assembly of short reads using OGs), we assemble all short reads whose alignments were not in full agreement with their long read contigs (blue in Fig. 3) or which had remained entirely unaligned with long read contigs (gray in Fig. 3). See the bottom of Fig. 3 for an illustration of the resulting assemblies.

As per the properties of StrainXpress, the result are strain-aware short read based contigs. As per the protocol we follow, all contigs refer to strain-aware metagenomic sequence not captured / spanned by any of the long read contigs from the first module.

**Third module: merging long and short read assemblies.** The purpose of this final module is to compute a unifying assembly that is as comprehensive as possible, as the final output of our approach.

1. One collects both long and short read contigs, as the output of the first and second module, and computes overlaps between them to establish an encompassing strain-aware overlap graph.
2. One identifies nodes in the OG through which only one particular path passes ("simple path").
3. One extends contigs along the identified "simple paths". The resulting extended contigs are the final output.

**Data & experimental setup**
The experiments we discuss in the following refer to both simulated and real data.

The *synthetic data* we treat refer to scenarios that reflect different levels of complexity in terms of strain content. In particular, we deal with data sets reflecting 3 *Salmonella* strains, and further 20 (low complexity), 100 (medium complexity) and 210 (high complexity) strains from various bacterial species. Strains were retrieved from ([55], DESMAN); see Supplementary Data for full information on

**Table 1 | Benchmark results for assembly simulated PacBio CLR reads**

| Assembly | GF(%) | NGA50 | Indels/100 kbp | Mismatches/100 kbp | N/100 kbp | MC(%) |
|---|---|---|---|---|---|---|
| *3 Salmonella* | | | | | | |
| MetaPlatanus | 72.25 | 68613 | 20.56 | 324.99 | 2.00 | 3.15 |
| Unicycler | 70.92 | – | 109.42 | 1957.53 | 0.00 | 6.88 |
| OPERA-MS | 68.43 | 41134 | 115.57 | 559.08 | 0.18 | 7.69 |
| hybridSPAdes | 46.22 | – | 35.73 | 816.90 | 0.00 | 1.60 |
| HyLight | **96.03** | **351848** | 0.85 | **23.56** | 0.00 | 0.19 |
| Strainberry | – | – | – | – | – | – |
| StrainXpress | 90.99 | 2645 | **0.7** | 59.61 | 0 | **0.07** |
| 20 strains | | | | | | |
| MetaPlatanus | 70.33 | 46882 | 39.11 | 238.79 | 23.21 | 2.15 |
| Unicycler | 68.59 | 46894 | 36.44 | 262.88 | 0.00 | 1.44 |
| OPERA-MS | 66.25 | 40464 | 189.74 | 290.46 | 0.09 | 3.66 |
| hybridSPAdes | 62.01 | 13237 | 11.83 | 333.44 | 0.00 | 0.44 |
| HyLight | 91.76 | **139730** | 3.97 | 59.96 | 0.00 | 0.20 |
| Strainberry | 78.43 | 95338 | 8.48 | **26.26** | 19.60 | 2.30 |
| StrainXpress | **92.74** | 1753 | **1.18** | 44.8 | 0 | **0.04** |
| 100 strains | | | | | | |
| MetaPlatanus | 76.60 | 46937 | 36.80 | 407.42 | 41.46 | 2.05 |
| OPERA-MS | 68.61 | 25223 | 39.31 | 575.26 | 0.53 | 7.90 |
| Unicycler | – | – | – | – | – | – |
| hybridSPAdes | – | – | – | – | – | – |
| HyLight | **93.99** | **163296** | 16.87 | 75.95 | 0.00 | 0.93 |
| Strainberry | 79.25 | 65706 | 210.30 | 115.87 | 68.05 | 4.84 |
| StrainXpress | 91.79 | 2597 | **2.00** | **75.48** | 0 | **0.12** |
| 210 strains | | | | | | |
| OPERA-MS | 71.15 | 43045 | 185.34 | 483.57 | 0.13 | 7.15 |
| MetaPlatanus | – | – | – | – | – | – |
| Unicycler | – | – | – | – | – | – |
| hybridSPAdes | – | – | – | – | – | – |
| HyLight | **90.49** | **128015** | 24.59 | 66.41 | 0.00 | 1.63 |
| Strainberry | 78.56 | 81842 | 418.36 | 102.55 | 41.72 | 4.66 |
| StrainXpress | 89.11 | 1240 | **1.89** | 53.42 | 0 | **0.19** |

Indels/100 kbp: average number of insertion or deletion errors per 100,000 aligned bases. Mismatches/100 kbp = average number of mismatch errors per 100,000 aligned bases. Genome Fraction GF reflects how much of each of the strain-specific genomes is covered by contigs. N/100 kbp denotes the average number of uncalled bases (N's) per 100,000 bases in contigs. MC = fraction of misassembled contigs. In the same comparison group, the best-performing results are highlighted in bold to emphasize their significance.

strain composition of data sets. All data sets were simulated using CAMISIM[56], which reflects a state-of-the-art and widely popular choice for generating metagenome sequencing data sets. Further, we also consider 6 "strain-mixing spike-in" data sets, which reflect spiking simulated reads from *Salmonella* strains into real data. This creates a real data scenario for which ground truth (in form of simulated reads) is available. See the "Methods" section for full technical details.

The *real data* are two microbial communities that reflect the current standard in terms of available real, both TGS and NGS data with known ground truth. Both Bmock12 (a bacterial mock community) and NWC (a natural whey culture data set) have already been widely used in the evaluation of metagenome assembly approaches[12,13,51,57,58]. For both data sets, reference genomes, Illumina, PacBio CLR and ONT reads are readily available. See again the Methods section for full technical details.

To assess the performance of the hybrid strategy and the assemblies generated solely from high-quality HiFi reads, we created a mock community by mixing real sequencing data from three yeast strains. The three sequencing data sets were originally intended for evaluating different sequencing data platforms and assembly methods[59]. Consequently, these data sets contain PacBio HiFi reads, ONT reads, and NGS reads. In this study, they serve as an excellent basis for assessing the performance of HyLight, which runs on NGS and ONT reads, in comparison to Hifiasm-meta[60] and metaMDBG[61], which solely utilize PacBio HiFi reads for assembly.

### Benchmarked approaches
As discussed in the Introduction, the state-of-the-art when comparing hybrid metagenome assembly approaches that operate in a strain-aware manner, are Strainberry, as the leading approach to deal with only TGS data and StrainXpress, as the leading approach to only deal with NGS data. Hybrid metagenome assembly approaches that address strain awareness have not been presented before; here, we consider all state-of-the-art approaches to metagenome assembly that operate at the species level. These are HybridSPAdes[25], MetaPlatanus[28], Unicycler[26] and Opera-MS[27]. Last, we also compare HyLight with Hifiasm-meta and metaMDBG to assess differences in quality of HyLight's hybrid assemblies with PacBio HiFi only based assemblies, as generated by the current state-of-the-art assemblers.

### Note on metrics
In the following, we evaluate the performance in terms of metrics that are routinely computed by MetaQUAST V5.1.0rc1[62] and Merqury[63]. For MetaQuast, we particularly focus on "Genome Fraction" (GF), as a metric that refers to strain awareness (GF = 100.0 translates into full strain awareness), NGA50, as a metric deemed sufficiently reliable to measure contig contiguity, and (mismatch / indel) error rates as well as misassembled contig fraction (MC) to evaluate the quality of the contigs. Among the evaluation metrics of Merqury, the focus is primarily on "Completeness", which, similar to MetaQuast's Genome Fraction, reflects the proportion of the genome covered by the assembled contigs. Additionally, we pay attention to the error rates reported by Merqury. Importantly, note that Merqury, as a k-mer based tool introduces particular biases in its evaluation, which was noted earlier where it was found to favor k-mer based tools[64]. Also, here, the corresponding statistics appear statistically uncertain; obviously, evaluating experiments without a ground truth comes at a price, which is not surprising. See "Methods" for full details on MetaQUAST and Merqury.

### Note on classification of approaches
We recall that the state-of-the-art hybrid assembly approaches primarily target the accuracy and the length of the assemblies, but do not address strain awareness. Strainberry and StrainXpress do primarily target at strain awareness. As a trade-off, they suffer from more erroneous (Strainberry) or shorter (StrainXpress) assemblies due to the nature of the type of data they use as their input: Strainberry and StrainXpress only use TGS or NGS data, respectively. HyLight is the sole approach that addresses accuracy, contiguity, and strain awareness at the same time.

Here, because of the different primary goals of the approaches, we would like to avoid to compare prior hybrid assembly approaches with prior non-hybrid approaches that focus on strain awareness. Therefore, in the following, we first compare HyLight with the prior hybrid assembly approaches, and, subsequently, in separate paragraphs, present a comparison of HyLight with Strainberry and StrainXpress.

**Table 2 | Benchmark results for assembly real reads**

| Assembly | GF(%) | NGA50 | Indels/ 100kbp | Mismatches/ 100kbp | N/ 100kbp | MC(%) |
|---|---|---|---|---|---|---|
| **Bmock12 ONT** | | | | | | |
| MetaPlatanus | 95.37 | **789960** | 34.34 | 66.81 | 178.39 | 4.60 |
| OPERA-MS | 94.30 | 207591 | 20.78 | 78.01 | 0.04 | 8.38 |
| hybridSPAdes | – | – | – | – | – | – |
| Unicycler | – | – | – | – | – | – |
| HyLight | **99.77** | 281944 | **1.45** | **3.58** | 0.00 | 3.59 |
| Strainberry | 67.60 | 688598 | 705.34 | 264.57 | 3.57 | 11.66 |
| StrainXpress | 99.04 | 65743 | 34.17 | 41.4 | 19.57 | **0.78** |
| **Bmock12 PacBio** | | | | | | |
| MetaPlatanus | 94.26 | **809518** | 15.85 | 31.83 | 29.11 | **4.29** |
| OPERA-MS | 93.20 | 179729 | 39.65 | 52.19 | 0.03 | 8.60 |
| hybridSPAdes | – | – | – | – | – | – |
| Unicycler | – | – | – | – | – | – |
| HyLight | **98.57** | 123823 | **5.29** | **19.24** | 0.00 | 7.62 |
| Strainberry | 62.50 | 72377 | 272.41 | 33.48 | 21.30 | 22.17 |
| **NWCs ONT** | | | | | | |
| MetaPlatanus | 68.54 | 20673 | 260.28 | 248.95 | 2497.22 | 4.03 |
| OPERA-MS | 62.52 | 7656 | 239.54 | 256.18 | 0.38 | 41.54 |
| hybridSPAdes | 56.36 | – | **16.00** | 102.93 | 0.00 | 4.59 |
| Unicycler | 53.78 | 10687 | 117.12 | **68.69** | 0.00 | 34.15 |
| HyLight | **95.35** | 62800 | 30.45 | 174.89 | 0.00 | 9.37 |
| Strainberry | 91.69 | **141570** | 764.14 | 193.84 | 60.26 | 22.94 |
| StrainXpress | 75.47 | 1056 | 25.57 | 399.71 | 0 | **3.38** |
| **NWCs PacBio** | | | | | | |
| MetaPlatanus | 63.86 | 839 | 116.67 | 143.66 | 802.98 | 4.19 |
| OPERA-MS | 56.49 | – | 446.73 | 315.74 | 0.41 | 26.57 |
| hybridSPAdes | 54.83 | – | **22.53** | 124.98 | 0.00 | **3.57** |
| Unicycler | 46.96 | – | 36.14 | **57.18** | 0.00 | 11.06 |
| HyLight | **78.94** | **22388** | 84.42 | 219.74 | 0.00 | 4.27 |
| Strainberry | 43.03 | – | 246.94 | 111.47 | 0.65 | 22.22 |

Indels/100 kbp: average number of insertion or deletion errors per 100,000 aligned bases. Mismatches/100 kbp = average number of mismatch errors per 100,000 aligned bases. Genome Fraction GF reflects how much of each of the strain-specific genomes is covered by contigs. N/ 100 kbp denotes the average number of uncalled bases (N's) per 100,000 bases in contigs. MC = fraction of misassembled contigs. In the same comparison group, the best-performing results are highlighted in bold to emphasize their significance.

**Misassembled contig rate of strain aware assemblers.** The results of HyLight, Strainberry and StrainXpress with respect to misassembled contig rate (MC in Tables 1 & 2) remain very consistent across all datasets. To avoid redundancies when comparing HyLight with Strainberry and StrainXpress in terms of misassembled contig rate, we will not go into detail with respect to each of the data sets we run experiments on. As a general trend—which applies with no exception on any of the data sets—HyLight and StrainXpress considerably outperform Strainberry. For (the solely NGS based) StrainXpress, this can certainly be attributed to the reduced length of the contigs. For Strainberry, this can be attributed to being based on solely TGS data, which prevents the detection of misassemblies thanks to the accuracy of auxiliary NGS data.

**Experiments: synthetic data sets**
**3 Salmonella.** This data set contains simulated reads from three distinct strains of *Salmonella*. The average coverage for Illumina (NGS) and PacBio (TGS) reads is 20X and 10X, respectively, reflecting a low-coverage TGS data scenario in particular, as

intended. Despite the low number of strains (3), the high degree of similarity between them ensures that only approaches that are sufficiently strain aware are able to assemble them without confounding them. The data set serves as a test bed for evaluating basic properties of the benchmarked approaches. See "Methods for full details.

**Hybrid assembly approaches.** See Table 1 for corresponding results. HyLight outperforms all other hybrid assembly approaches in terms of all relevant metrics. It covers 23.78% more strain sequence than the second best hybrid assembly approach (HyLight: 96.03; MetaPlatanus: 72.25), missing out on only 4% strain-specific sequence. HyLight also dominates the other approaches in the other relevant categories, where improvements are to be measured in terms of orders of magnitude. For example, it improves NGA50 by a factor of 5 (HyLight: 351 848; MetaPlatanus: 68 613), indel error rate by a factor of 24 (HyLight: 0.85/100 kbp versus 20.56/100 kbp), mismatch error rate by a factor of 13.7 (HyLight: 23.56/100 kbp; MetaPlatanus: 324.99/100 kbp) and missambled contig rate (MC) by a factor of 8.4 over the second best approach (HyLight: 0.19%; MetaPlatanus: 1.6%).

**Strainberry / StrainXpress.** See Table 1. Strainberry encountered difficulties in the phasing step due to the high similarity among the three *salmonella* strains (ANI >99%). As a result, Strainberry could not identify sufficiently many SNPs to separate reads from contigs, as assembled by Metaflye. Consequently, Strainberry was unable to assemble this dataset. StrainXpress achieves Genome Fraction that is superior over all prior (including hybrid) approaches, but outperformed by HyLight (90.99%), and achieves excellent results in categories relating to contig quality (errors and misassemblies). However, in terms of contiguity, StrainXpress lags behind all other approaches, by large margins. This is no surprise, of course, since StrainXpress is the only approach that does not make use of long reads, which limits its potential to output longer contigs.

**20 bacterial strains.** This data set consists of 20 strains from 10 different species, resulting in an average of two strains per species. The average coverage for Illumina (NGS) and PacBio (TGS) reads is 20X and 10X, respectively, again reflecting a TGS low coverage scenario. For further details, please refer to the "Methods" section.

**Hybrid assembly approaches.** See again Table 1. Also on this data set, HyLight outperforms the other four methods across all categories. HyLight achieves a Genome fraction of 91.76%, surpassing the current best method by more than 21% (MetaPlatanus: 70.33%). The NGA50 of HyLight is 139,730, which exceeds the second best NGA50 by a factor of 3 (Unicycler: 46 894). Regarding errors, HyLight's contigs mark a threefold improvement in terms of indel error rates (HyLight: 3.97/ 100 kbp; HybridSPAdes: 11.83/100 kbp), and the mismatch error rate is four times lower than the toughest competitor (HyLight: 59.96/ 100 kbp; MetaPlatanus: 238.79/100 kbp). Finally, there are only half as many misassembled contigs relative to the second best competing method (HyLight: 0.20%; HybridSPAdes: 0.44%).

**Strainberry / StrainXpress.** As was expected, both Strainberry and StrainXpress are competitive with respect to Genome Fraction. However, while StrainXpress (93.45%) even outperforms HyLight (91.76%), Strainberry achieves only 78.43%, which from an overall perspective (including hybrid approaches) still is remarkable. In terms of contiguity, unlike Strainberry's assembly, whose contiguity is worse than that of HyLight, but still competitive, StrainXpress' NGA50 is smaller by a factor of more than 40 in comparison with HyLight. Further, Strainberry's error rates are largely on par with the low error rates of HyLight and StrainXpress, which is somewhat surprising in particular

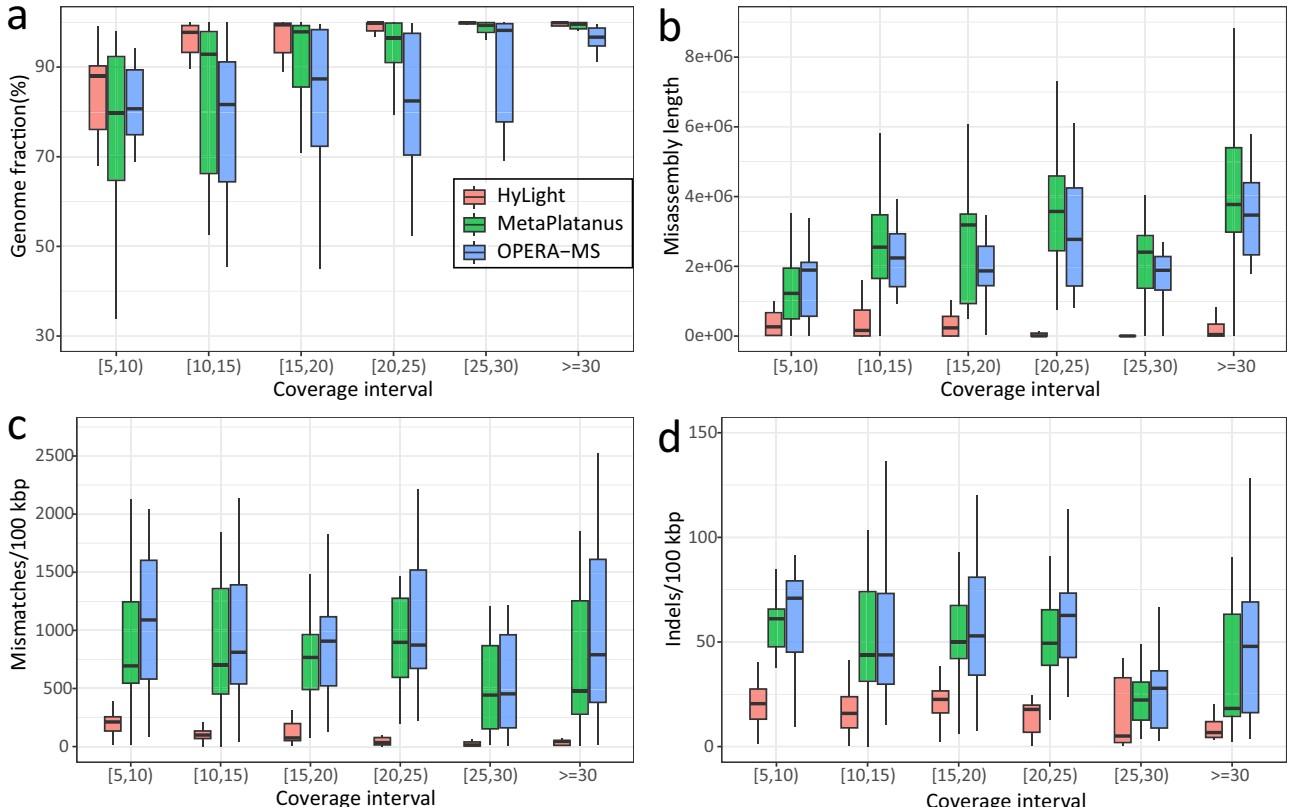

**Fig. 4 | Assemble 100 strains.** Among these 100 strains, their average coverages follow a log-normal distribution, resulting in variations in the average coverage of each individual strain. Here, we assess the impact of different coverages on the assembly methods of HyLight, MetaPlatanus, and OPERA-MS. Different colors represent different assembly approach. (**a**) As coverage increases, there is a change in the genome fraction for distinct approaches. (**b**) As coverage increases, there is a variation in misassembly contig length among different assembly methods. **c** and **d** Increase in coverage, changes in mismatch and indel error rate in the assembly results of different approaches. Box plots represent the median (center line), the 25th and 75th percentiles (bounds of the box), and the minimum and maximum values within 1.5 times the interquartile range (whiskers).

for indel errors. While the good error rates of Strainberry can be attributed to the low complexity of the data set, results in the other categories reflect expected outcomes when working with low coverage TGS and/or (medium coverage) NGS data. Here, just as much as on all other data sets, StrainXpress has the lowest misassembled contig rate (0.04% vs. 0.20% by HyLight).

**100 bacterial strains.** This data set consists of 100 strains from 30 species, at an average coverage of 20X per strain for NGS (Illumina) and the (as usual low) 10X per strain for TGS (PacBio CLR) reads. The data set is designed to reflect a more complex scenario. The idea is to evaluate which of the available approaches potentially become confused if the mix of strains becomes more complex and more diverse.

**Hybrid assembly approaches.** See Table 1). In fact, despite a slight increase in terms of errors, HyLight remains unaffected by the elevated complexity and continues to outperform the other four methods. Note first that neither HybridSPAdes nor Unicycler was able to perform the assembly within a month time, so we terminated the corresponding runs (on 32 CPUs and 500 GB RAM) not terminating when the strain number reached 100, short read volume reached 16G, and long read volume reached 10G). As for Genome Fraction, HyLight outperforms the other methods by at least 17%, where GF even exceeds the GF achieved on the low complexity data set (HyLight: 93.99%; MetaPlatanus: 76.6%). The NGA50 exceeds the second best one by 3.5 times (HyLight: 163 296; MetaPlatanus: 46 937). Indel error rates are still lower by a factor of more than 2 (HyLight: 16.87/100 kbp;

MetaPlatanus: 36.8/100 kbp) and mismatch error rates are smaller by a factor of more than 5 (HyLight: 75.95/100 kbp; MetaPlatanus: 407.42/100 kbp). Misassembled contig rate is smaller by a factor of more than 2 (HyLight: 0.93; Metaplatanus: 2.05).

Thanks to variations in the average coverage of the TGS data of the 100 strains (average coverage follows a log-normal distribution by the design of CAMI), one can analyze the influence of coverage on the quality of the assemblies of the different strains. See Fig. 4 for the corresponding results. In comparison to the other two methods whose runs terminated successfully, HyLight generally achieves greater Genome Fraction across all strains. HyLight's advantages become particularly noticeable at coverage rates below 20X where HyLight outperforms the other methods by large margins with respect to all categories that refer to strain awareness and error content.

**Strainberry / StrainXpress.** Again, StrainXpress excels in terms of strain awareness (Genome Fraction), closely followed by HyLight. Although the margin between Strainberry and HyLight is considerable, Strainberry still achieves remarkable strain awareness with from an overall perspective that takes the other hybrid approaches into account. The increased complexity of the data has an impact on contiguity and error rates of the assemblies. Strainberry now has considerable disadvantages in terms of error rates, in particular with respect to indel errors, while HyLight and StrainXpress preserve excellent error rates. StrainXpress has considerable disadvantages in terms of contiguity, where HyLight exceeds the NGA50 of Strainberry by more than 2 times.

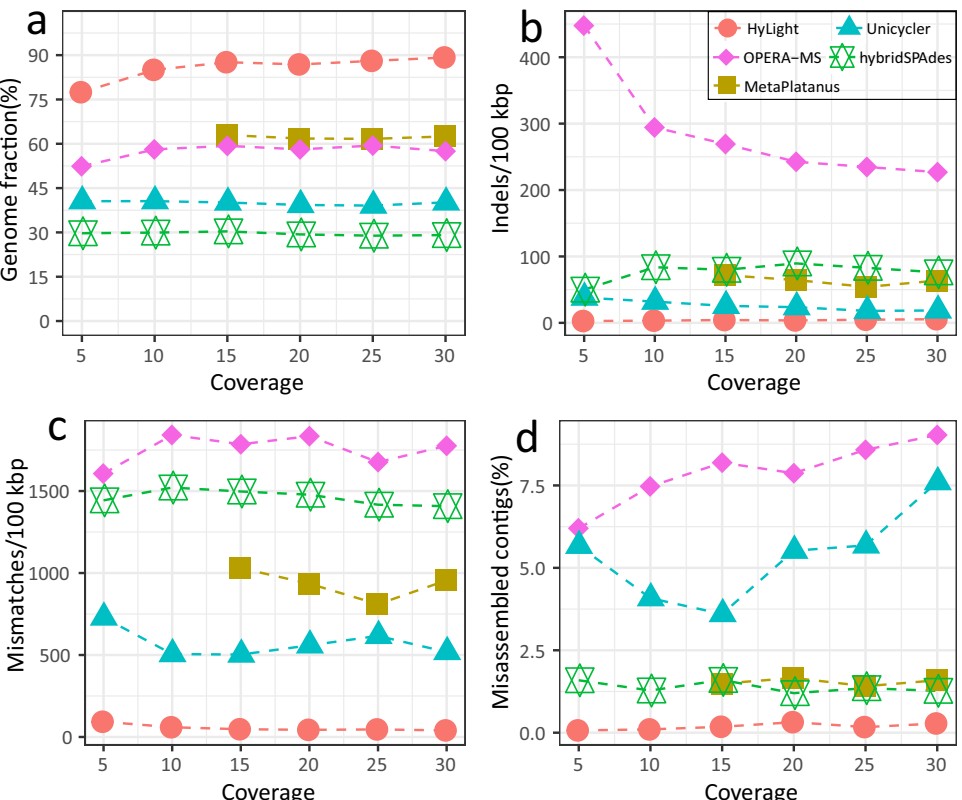

**Fig. 5 | 10 *Salmonella* strains spike-in.** Simulated reads of 10 *Salmonella* strains were mixed with real sequencing data, followed by assembly for the combined datasets. Subsequently, the quality of assembly results for these spike-in strains in a complex environment was evaluated. During the incremental increase of coverage from 5X to 30X, the variations in genome fraction (**a**), indel error rate (**b**), mismatch error rate (**c**), and misassembly contig rate (**d**) were presented for these five assembly methods.

**210 bacterial strains.** This data set consists of 210 strains from 100 species as provided by[55], for which reads were simulated using CAMISIM[56]. As usual, depth of coverage is 10X for TGS (so low coverage), and 20X for NGS reads. The data set is supposed to reflect a scenario of utmost complexity with respect to numbers of species and their strains.

**Hybrid assembly approaches.** See Table 1 for results. By and large, HyLight, as well as Opera-MS, as the only methods whose runs terminated within a month time, approximately mirror the results achieved on the data set containing 100 strains. HyLight outperforms Opera-MS by 19% in terms of Genome Fraction (HyLight: 90.49%; Opera-MS: 71.15%), has three times longer contigs (NGA 50: HyLight: 128 015; Opera-MS: 43 045), has more than 3 times lower indel error rates (HyLight: 24.59/100 kbp; Opera-MS: 185,34/100 kbp), 7 times lower mismatch error rates (HyLight: 66.41/100 kbp; Opera-MS: 483.57/100 kbp) and more than 4 times less misassembled contigs (MC: HyLight: 1.63%; Opera-MS: 7.15%).

**Strainberry / StrainXpress.** Results largely repeat the achievements from the data set on 100 strains just discussed, but become even more distinct in terms of the expected advantages and disadvantages of the approaches. Both approaches outperform other approaches in terms of strain awareness. On this most complex data set, HyLight finally also clearly outperforms StrainXpress. While Strainberry has drawbacks with respect to error rates, StrainXpress considerably trails in terms of contiguity, with HyLight clearly outperforming both approaches in these categories.

**Strain-mixing spike-in datasets.** By its design, these data sets can be used to investigate the influence of the coverage of the NGS reads in hybrid assembly. To enable such experiments, we made use of 10 highly identical *Salmonella* strains, which we spiked into real metagenome samples. While the coverage of spiked-in long reads was fixed to 10X, the coverage of spiked-in NGS reads varied from 5X to 30X, in steps of 5X, resulting in 6 different levels of coverage. These 6 different NGS read sets were spiked into 6 different real metagenome sequencing data sets, amounting to 36 different data sets overall. For each of these 36 data sets, the task is to assemble the genomes of the 10 *Salmonella* strains, in a strain-aware manner.

For the evaluation, note that we do not make use of MetaQuast, because MetaQuast was not able to align the contigs against the reference effectively due to the high identity of many strains, often amounting to average nucleotide identity (ANI) of more than 99%. This implied that indel and mismatch error rates were evaluated as excessive for all methods apart from HyLight (for example, the mismatch error rate was evaluated as 6859.95/100 kbp for Opera-MS, which cannot be correct, see Supplementary Data). For fairness reasons, we therefore resorted to using Quast[65] with the same parameters, because we realized that Quast aligned contigs with the reference genomes more accurately. This immediately entailed that the error rate of the competitors dropped substantially (e.g., Opera-MS now at 1753.41/100 kbp). See Supplementary Data for both Quast and MetaQuast evaluated results.

An additional challenge was that MetaPlatanus consistently threw errors when dealing with data sets of only 5X or 10X simulated NGS coverage. Despite reaching out to the authors (via GitHub), the issue

could not be resolved. Therefore, we only display results for datasets of simulated NGS coverage 15X and greater.

**Hybrid assembly appraoches.** See the Fig. 5 for results. HyLight outperforms the other methods across different coverages. For example, HyLight achieves an average Genome Fraction that exceeds those of other approaches by at least 28.81% (24.65% ~ 26.93%). Note that at coverage 5X, the Genome Fraction of HyLight drops to 77.45%. Genome Fraction for HyLight already increases to 85.03% when increasing coverage to 10X. Increasing coverage further does not lead to any more significant changes. Subsequent increases in the coverage of the most community did not have a more significant changes (Genome Fraction rises to nearly 90% from 15X and onwards).

Among the prior hybrid assemblers, Unicycler achieves the lowest indel error rate (17.97/100 kbp ~37.98/100 kbp) where HyLight (2.73 ~ 5.47/100 kbp) achieves an error rate of only 17.5% of that of Unicycler (Fig. 5b). Improvements of HyLight over prior hybrid assemblers in terms of mismatch errors are even more distinct, decreasing the one of the top competitor by about 90%. Figure 5d finally displays comparison in terms of misassembled contig rate (MC): HyLight's MC is only 17.5% of that of HybridSPAdes, as the toughest competitor.

In summary, HyLight outperforms the state-of-the-art in hybrid assembly by large margins with respect to the most relevant key assembly metrics, in a scenario that is characterized by near-identical strains embedded into complex real backgrounds.

*Strainberry / StrainXpress* are not evaluated, because the experiment only makes sense for evaluating particular qualities of hybrid assembly approaches.

## Experiments: real data sets

We conducted further evaluations of all approaches using four real datasets: "Bmock 12 PacBio", "Bmock 12 ONT", "NWCs PacBio" and "NWCs ONT". While TGS coverage of the 4 data sets amounted to 22.11X, 18.1X, 127.2X and 89.01X (in the order of having listed data sets before), NGS coverages reached 275X for Bmock12 and 35.62X for NWCs.

The 'Bmock12' dataset consists of 11 strains from 9 species. Due to the low number of strains per species, also strain-unaware approaches are to achieve fairly high Genome Fraction overall. This also means that a sufficiently thorough analysis of the strain awareness of the approaches requires to break down results relative to the strains that make part of the data set. In this vein, one notices that among the species present, only *Marinobacter* and *Halomonas* have more than one strain, see Table 3 for summarizing statistics that refer to "Bmock 12 ONT" (statistics for "Bmock 12 PacBio" are similar, see Supplemen-

tary Table 1). While the two strains of *Marinobacter* exhibit 85% average nucleotide identity (ANI), the two *Halomonas* strains have an ANI of 99%. This points out that methods should be evaluated with a view to their performance on *Halomonas* strains in particular.

The NWC dataset includes 3 species (*Streptococcus thermophilus, Lactobacillus delbrueckii, Lactobacillus helveticus*), each of which has 2 strains, at ANIs of 99.99%, 99.24%, and 98.03%, respectively. For more detailed information, please see Supplementary Table 3. Despite the limited number of strains and their relatively low complexity, assembly remains challenging due to the high degree of similarity affecting the two strains of a particular species.

## Bmock12 ONT

**Hybrid assembly approaches.** See Table 2 for the following results. Due to the reduced level of complexity in terms of the variety of strains, also strain-unaware, species-level metagenome assembly approaches are expected to deliver good performance in reconstructing the individual genomes. Despite the reductions in overall coverage due to the subsampling procedure (inducing a low coverage TGS data scenario), both HybridSPAdes and Unicycler were unable to complete the assembly process within one month runtime.

HyLight outperforms both Opera-MS and MetaPlatanus, which are the two hybrid assembly approaches whose runs terminate in acceptable time when examining the relevant criteria and putting them into mutual perspective. Genome Fraction of HyLight is 4.4% higher than that of the second-ranked MetaPlatanus (99.77% vs. 95.37%). MetaPlatanus achieves the greatest NGA50 (789,960 vs. 281,944), which, however, can be explained by the unusually large number of Ns in its contigs, whose primary purpose is to link and extend contigs by force without that read evidence for the missing sequence context of contig links can be provided. HyLight improves by more than one order of magnitude over the other methods in terms of indel and mismatch errors. The indel error rate of HyLight is only 6.9% of that of the second-ranked OPERA-MS (1.43 vs. 20.78 per 100 kbp), and the mismatch error rate of HyLight is only 5.4% of that of the second-ranked MetaPlatanus (3.58 vs. 66.81 per 100 kbp).

We further examined the assembly status for each strain making part of the mock community individually. Genome Fraction for the individual strains is displayed in Table 3. One immediately realizes that all approaches reconstruct at least (about) 99% of the strain-specific sequence for all but the two *Halomonas* strains whose ANI comes at 99%. On *Halomonas sp.HL-4* in particular, HyLight achieves a Genome Fraction that is greater by 7.58% than that of MetaPlatanus and 8.93% than that of Opera-MS (HyLight: 99.36; MetaPlatanus: 93.05; Opera-MS: 90.7). This lets one conclude that HyLight is the only hybrid

## Table 3 | The genome fraction of each individual strain in the Bmock12 data (Illumina and ONT)

| Assembly | Coverage (Illumina) | Coverage (ONT) | HyLight | MetaPlatanus | OPERA-MS |
|---|---|---|---|---|---|
| **Halomonas sp.HL-4** | **507.08** | **46.03** | **99.63** | **93.05** | **90.70** |
| **Halomonas sp.HL-93** | **579.87** | **41.23** | **99.68** | **96.75** | **93.90** |
| **Marinobacter sp.LV10R510-8** | **447.83** | **18.19** | **100.00** | **99.97** | **99.80** |
| **Marinobacter sp.LV10MA510-1** | **135.05** | **6.95** | **99.94** | **99.85** | **98.95** |
| Muricauda sp.ES.050 | 618.76 | 53.87 | 100.00 | 99.92 | 99.82 |
| Psychrobacter sp.LV10R520-6 | 425.47 | 31.82 | 100.00 | 99.34 | 98.46 |
| Cohaesibacter sp.ES.047 | 170.59 | 11.78 | 99.78 | 99.48 | 98.57 |
| Thioclava sp.ES.032 | 78.32 | 5.54 | 99.64 | 99.65 | 99.40 |
| Propionibacteriaceae bacterium | 31.90 | 2.39 | 99.99 | 99.99 | 99.97 |
| Micromonospora echinofusca | 18.19 | 2.44 | 99.58 | 99.50 | 99.29 |
| Micromonospora echinaurantiaca | 14.91 | 1.80 | 99.64 | 99.67 | 99.11 |

Present the impact of different sequencing coverage in different assembly methods. Because these are the only four strains in Bmock12, their assembly is more challenging than others, so they are highlighted in bold to emphasize their importance.

metagenome assembly approach that operates in a strain-aware manner even when the ANI between strains is as great as 99%. As an additional insight gained from the analysis of the quality of the assemblies that is stratified relative to the individual strains, one realizes that all hybrid assembly approaches are able to reconstruct about 99% of the strain specific sequence even when TGS sequencing coverage is as low as 1.8X and also as low as 14.91X for the respective NGS coverage for a particular strain (here: *Micromonospora echinaurantiaca*). This documents the general value of hybrid metagenome assembly with respect to its favorable behavior in terms of sequencing (in particular TGS) coverage demands.

**Strainberry / StrainXpress.** While StrainXpress achieves competitive performance with respect to Genome Fraction (99.04%), Strainberry somewhat unexpectedly, no longer does (67.60%), while HyLight outperforms also StrainXpress (99.77%). While Strainberry has drawbacks with respect to error rates, StrainXpress considerably trails in terms of contiguity. HyLight outperforms both approaches in terms of both indel error and mismatch error rates, but is outperformed by Strainberry in terms of contiguity.

**Bmock12 PacBio.** Note that NGS reads used here agree with those from "Bmock12 ONT", whereas the TGS reads now stem from PacBio CLR sequencing platforms. This means in particular that the results of StrainXpress (see further below agree with those achieved on "Bmock12 ONT".

**Hybrid assembly approaches.** See Table 2 for results. In an overall summary, results here mirror results achieved on "Bmock12 ONT", where the advantages of HyLight look less: HyLight maintains a Genome Fraction that exceeds that of the other approaches by more than 4%, and although less distinct, still exhibits considerably lower error rates. The NGA50 of MetaPlatanus exceeds that of HyLight considerably, again put into context by the large number of N's in the MetaPlatanus contigs. Misassembled contig rate of HyLight is again roughly on a par with that of MetaPlatanus, where now MetaPlatanus has slight advantages (but see below for a more fine-grained analysis that provides explanations). Although results look more favorable for MetaPlatanus from a greater persepctive, breaking down results by strain (see Supplementary Table 2) reveals that MetaPlatanus substantially struggles in reconstructing one of the *Halomonas* strains: while achieving 93.05 Genome Fraction for *Halomonas sp.HL-4* on ONT data, MetaPlatanus only achieves 82.19 Genome Fraction on PacBio data, despite even having small advantages over HyLight on the other *Halomonas* strain. Analogous trends become evident for Opera-MS. With respect to the misassembly contig rate, a more detailed analysis further demonstrates that the MC of the raw long reads (i.e., evaluating raw long reads as contigs in their own right) for "Bmock12 ONT" and "Bmock12 PacBio" comes out at 2.33% and 7.55%, see Supplementary Table 1. The MC of the raw long reads are introduced by chimera reads. Comparing the MC of the raw long reads with the MC of HyLight points out that HyLight reproduces MC rates of the raw long reads. The most plausible explanation for this is the fact that HyLight uses overlap graphs, which cannot identify chimera reads, while "short-read-first" approaches, thanks to employing DBG's, can identify artificial links as mistaken. While there is good hope that overlap graph based approaches are able to identify chimera reads, too, we leave such improvements as promising future work at this point.

**Stainberry / StrainXpress.** StrainXpress reproduces its results by making use of only the NGS portion of the data, which agrees with that from "Bmock12 ONT". Again, Strainberry, somewhat unexpectedly, does not achieve competitive performance in terms of strain awareness. Similarly, Strainberry again has considerable drawbacks with respect to (in particular indel) error rates, containing more then 40

times more indel errors than HyLight. Here, also the contiguity of Strainberry's assembly here is outperformed by that of HyLight (NGA50 - HyLight: 123823; Strainberry: 72377).

### NWC ONT

**Hybrid assembly approaches.** Due to the presence of a higher number of highly similar strains in NWC compared to Bmock12, the advantage of HyLight becomes more pronounced. HyLight outperforms the other hybrid assembly methods, both in terms of Genome Fraction (i.e., strain awareness; HyLight: 95.35%; MetaPlatanus, as second best: 68.54%) and NGA50 (i.e., contiguity; HyLight: 62800; MetaPlatanus, as second best: 20673). Further, although not outperforming the other approaches, HyLight achieves decent indel and mismatch error rates in comparison with the other approaches, ranking second and third, respectively, and, just like most other approaches, reducing the error content of the raw long reads by more than two orders of magnitude. Again, the MC, although not bad, is slightly worse than that of MetaPlatanus and HybridSPAdes, again reflecting that HyLight adopts issues introduced by chimera reads, for which there is good hope that this can be successfully addressed in future work.

**Strainberry / StrainXpress.** All approaches achieve competitive performance relative to strain awareness, in case of StrainXpress at least from the perspective of comparing it with strain unaware approaches, as was expected, while HyLight, followed by Strainberry both excel. Strainberry has drawbacks with respect to indel error rates, containing approximately 25 times more indel errors than HyLight, while being roughly on a par with HyLight in terms of mismatch errors. As usual, StrainXpress considerably trails in terms of contiguity, with Strainberry taking over the lead from HyLight.

**NWC PacBio.** Unlike NWC ONT, this data set is affected by two strains whose long read coverage is extremely low (1.45X and 2.47X, respectively see Supplementary Table 5). The reconstruction of those two strains presents a particular challenge, which points out that this data set is a particularly challenge in one overall. Therefore, all methods produced assembly results that were inferior to those achieved on NWC ONT.

**Hybrid assembly approaches.** Notwithstanding the level of difficulty of the data set overall, results virtually reproduce the ones achieved on NWC ONT: HyLight outperforms the other approaches both in terms of Genome Fraction (HyLight: 78.94%; MetaPlatanus: 63.86% as second best) and NGA50 (HyLight: 22388; MetaPlatanus: 839 as second best). Error rates are roughly on a par with those of the other approaches, where everyone achieves sufficiently decent results. MC is slightly lower than those of the two best approaches, but considerably better than those of the other two approaches; again, presumably, chimera reads imply that HyLight reproduces the MC rates of the raw reads.

**Strainberry / StrainXpress.** Just as for Bmock12, StrainXpress reproduces its results because making use of only the NGS portion of the NWC data. Again, Strainberry's performance in terms of strain awareness drops (here: quite substantially), which is somewhat unexpected, and may be due to reduced quality of the TGS portion of the data here—note that even HyLight somewhat struggles. In comparison to "NWC ONT", Strainberry's error rates are improved, containing only approximately 3 times more indel errors than HyLight, with StrainXpress taking the lead in terms of indel errors. Mismatch error rates are similar to those of NWC ONT. In terms of contiguity, HyLight clearly outperforms Strainberry, whose contigs only align with less than 50% of the true genome, which prevents computation of NGA50.

**Hybrid versus HiFi: three yeast strains.** The three yeast strains *Saccharomyces cerevisiae* S288C, CICC-1445, and *Saccharomyces pombe* FLO-DUT were sequenced with PacBio HiFi, Oxford Nanopore Technologies, and the short-read sequencing technology BGISEQ (2 × ! 150 bp paired reads), which allows to evaluate experiments that compare hybrid approaches (as per their design relying on ONT and BGISEQ) with HiFi assemblers. To control variables, and make sure that we were dealing with low coverage datasets, we subsampled 10X data from each sequencing technology for the assemblies. It is noteworthy that the ONT reads here contained a large number of shorter reads, resulting in an average length only half of that of PacBio HiFi reads, which is not the typical case. Generally, ONT reads are two to three times the length of PacBio HiFi reads. To ensure fairness in subsequent evaluations, we randomly subsampled reads with the requirement that ONT reads should be longer than 10,000 bp and PacBio HiFi reads should be longer than 5,000 bp. Due to the abundance of relatively shorter reads in the ONT data for these three datasets, the resulting 10X ONT subset had an average length close to that of PacBio HiFi.

Subsequently, to mimic a scenario akin to a metagenome, we merged the ONT, the BGISEQ and the HiFi reads of the three yeast strains into one data set for each sequencing technology. This established a mock community that consisted of two *S.cerevisiae* strains (S288C and CICC-1445) as well as a *S.pombe* strain (FLO-DUT). We ran all assemblers on this mock community type data set. Note that only S288C comes with a (haploid) reference genome (provided via the SRA), in other words only S288C is equipped with a ground truth. To evaluate results in a way that agrees with the conventions of evaluating metagenomes, we fed the contigs of each method into Merqury. We recall that the lack of reference genomes for CICC-1445 and FLO-DUT prevented an evaluation with MetaQUAST.

Results are shown in Table 4. Evidently, HyLight outperforms both HiFi assemblers, Hifiasm-meta and MetaMDBG quite substantially, while Hifiasm-meta and MetaMDBG are largely on a par. In summary, the hybrid assembler HyLight proves to be superior over the HiFi only assemblers. Evidently, combining noisy ONT with NGS data appears to be quite preferable over assemblies generated from HiFi data alone: the hybrid approach excels all in terms of strain awareness (Completeness), error content (Error Rate) and potential misassembled contigs (QV).

To make sure that these results agree with what one can achieve in terms of evaluating results with respect to an available reference genome, we also ran the contigs pertaining to S288C against its available reference genome. See Supplementary Table 8 for the corresponding results. Numbers confirm the superiority of HyLight's hybrid assemblies over the HiFi only assemblies all in terms of strain awareness, misassembly and error content. One can also see that the contiguity of the HiFi only assemblies exceeds that of HyLight. This suggests that HiFi assemblies trade length for quality in terms of strain specificity, error and misassembly content. However, since corresponding related results for the other two strains cannot be obtained because of the lack of reference genomes, one cannot be certain about the contiguity of the contigs relating to these two strains, since Merqury can only assess strain awareness and quality in terms of error and misassembly content.

**Runtime and memory usage evaluation**

We evaluated the performance of runtime and peak memory of all methods on the data set containing the 3 *Salmonella* strains, on a x86_64 GNU/Linux machine with 48 CPUs. The data volume of the NGS (Illumina) reads amounted to 573 MB and the volume of the Pacbio CLR reads amounted to 281 MB. Supplementary Table 7 reports CPU times and peak memory usages of the different hybrid assembly methods. without any doubt, OPERA-MS is the fastest tool: it only takes 2.09 hours and 1.23 GB memory. The runtime of hybridSPAdes, HyLight, and MetaPlatanus is roughly on a par, requiring

approximately 5.53, 7.01, and 6.93 hours, respectively. However, in terms of peak memory usage, both hybridSPAdes and HyLight demonstrate significantly lower usage compared to MetaPlatanus, with values of 3.85, 15.99, and 69.26, respectively. Unicycler, on the other hand, requires the longest runtime (53.71 hours).

## Discussion

Despite exhibiting a high degree of similarity, different strains of the same bacterial species can vary significantly in terms of phenotype, such as drug resistance or pathogenicity. This explains why it is important in biomedical and clinical research to identify putative pathogens at the level of strains as the desirable degree of taxonomic resolution, and not only at the level of species. Therefore, when analyzing an environmental mix of genomes, the current driving challenge is to assemble the individual genomes at strain resolution. In this, de novo assembly is crucial to avoid reference induced biases that would favor the detection of more common strains over those not yet fully investigated or even entirely unknown.

In this paper, we have presented HyLight, as an approach that aims at pushing the limits of strain-aware metagenome assembly in various aspects. HyLight is based on de novo hybrid assembly, characterized by integrating both long, third-generation sequencing reads and short, next-generation sequencing reads during the assembly process. Two good reasons support the principled superiority of hybrid assembly.

### The first reason is the superiority of the assemblies themselves

HyLight improves the quality of strain-level metagenome assemblies in comparison with the state of the art all in terms of strain awareness, contiguity, and accuracy (as measured by error and misassembly content), often by large margins. As anticipated, HyLight achieves its most pronounced advantages when strains are very similar and/or when strains are subject to low (TGS) read coverage. HyLight stands out as the sole method that consistently reconstructs at least 90% of the strain-specific sequence content, with second best approaches achieving only approximately 70% on average. Another particular advantage of HyLight is the low error content in terms of insertions and deletions, as the predominant type of errors that affects TGS reads. Beyond clearly outperforming competitors in terms of indel error content, HyLight shares the favorable non-indel error rates that the other approaches had been able to achieve already, which documents the beneficial complementarity of the TGS reads on the one hand and NGS reads on the other hand.

### The second reason is the substantial savings in terms of costs and resources

From the point of view of the global assembly/sequencing community, hybrid assembly opens up a range of opportunities for laboratories that operate on less generous budgets and are equipped with earlier types of TGS and NGS platforms, which applies for the great majority of laboratories worldwide. The principled explanation for this is the favorable complementarity of the two types of data: while TGS reads are long but inaccurate, and in particular predominantly affected by indel errors, NGS reads are short and accurate where the (anyway little) errors come in form of single letter mismatches. This reduces the need for generating larger data volumes. In line with earlier hybrid assembly approaches, HyLight again demonstrates that superior results are achieved already with little and cheap data.

It seems to be a fair assessment to consider most advanced types of reads (such as PacBio HiFi or ONT Q30+) as pleasant convenience that, unfortunately, remains unaffordable to the (vast) majority of laboratories worldwide. Beyond the expenses that these most advanced platforms require, our results show that also the quality of their assemblies is exceeded by that of our hybrid assemblies. One of the reasons is the length of the TGS reads of the earlier types, which

**Table 4 | Benchmark results are presented for the assembly of a real sequencing dataset comprising reads from 3 yeast strains**

| Assembly | Completeness (%) | QV | Error rate (%) |
|---|---|---|---|
| HyLight (Hybrid) | **98.08** | **41.55** | **0.007** |
| Hifiasm-meta (HiFi) | 96.31 | 30.94 | 0.081 |
| MetaMDBG (HiFi) | 93.49 | 33.31 | 0.047 |

Completeness indicates the fraction of the genomes covered by the assembled contigs. QV is a k-mer-based metric quantifying the quality of the assembled contigs. The error rate represents the proportion of erroneous bases present in the assembled contigs. In the same comparison group, the best-performing results are highlighted in bold to emphasize their significance.

exceeds the length of PacBio HiFi for example by factors of 2-4 times on average. Another reason may be the complementarity of NGS and TGS reads in terms of error content. Our results demonstrated again that using NGS reads for correcting errors in TGS reads results in ultimately low error content of the TGS reads or their contigs.

In summary, we have presented an approach that leverages the complementarity of ultra-long, but erroneous TGS reads and short, but accurate NGS reads. We have demonstrated to outperform the state of the art often by large margins, and on low data volumes in particular, which confirms the theoretical intuition of the approach. From the point of view of costs, we have indeed opened up opportunities for the many laboratories that operate on less generous budgets when it comes to the strain aware assembly of metagenomes.

The methodical key to success was to considerably remodel the protocols of hybrid assembly approaches. Unlike all of the earlier hybrid assembly approaches, HyLight does not follow a "short-read-first" or "long-read-first" protocol. Rather, HyLight employs a protocol that relies on assembling both types of reads without the protocol becoming too complex, which one could refer to as a "cross hybrid" or "mutual support" strategy. The foundation of this has been to use overlap graphs instead of de Bruijn graphs, where the latter have been dominating hybrid metagenome assembly approaches for several years. The crucial insight is the fact that using overlap graphs leads to the correct line up of strain specific variation across strain specific genomes, which is favorable even when working with short reads. Based on such accurate line ups, HyLight incorporates a filtering step that identifies mistaken (i.e., "strain unaware") overlaps among contigs, and removes them from further consideration, which constitutes an important technical advance.

In conclusion, we have introduced HyLight, as a de novo hybrid metagenome assembly approach that reconstructs the genomes of the individual members of microbial communities at strain level, which, to the best of our knowledge, is a novelty. HyLight is characterized by its economic behavior in terms of costs and times, and the ubiquitous availability of the data that HyLight relies on. This may enable a large number of laboratories worldwide that had been restricted to operate at the level of species, to now perform strain aware analyzes of environmental mixes of microbes.

Despite the various advantages that we have been able to demonstrate, there is still room for improvement. So far, for example, we have not yet addressed the existence of long chimera reads, which introduce a detectable amount of misassemblies and confound the correct identification of strain specific genomes to a small, but non-negligible degree. Last but not least, although HyLight's runtimes are perfectly viable, overlap graphs remain "heavy" data structures. In future work, we will focus on the design of "lightweight" overlap graphs that, although agreeing on minor amounts of inaccuracies, still achieve superior results in compensation for further substantial savings in terms of runtime and peak memory requirements.

## Methods

### Quality control

Before assembling reads with HyLight, we performed quality control on the sequencing reads using *fastp* (version 0.20.1)[66]. This multi-functional FASTQ data preprocessing toolkit ensures the quality of the data by providing major functions including quality control, adapter detection, base correction, and read filtering. In the raw reads, bases with Phred scores less than 20 at the 5' or 3' ends, as well as adapters, were trimmed. After trimming, only reads longer than 70 bp were retained. Moreover, in the overlapped regions of paired-end reads, *fastp* corrected mismatched bases only when a high-quality base was paired with a low-quality base.

### Workflow: detailed description of 3 modules

#### Module 1: Strain-Aware Assembly of Long Reads

**Hybrid correct long reads and establish OG.** As long-read technologies such as CLR or ONT often contain a significant amount of sequencing errors, which can affect downstream SNP identification and assembly, the first step in our pipeline is to use FMLRC2[49] to perform error correction on long reads using high-quality short reads. After obtaining high-quality long reads, we use minimap2 to align them to each other and construct an OG. In this OG, each read is represented as a node, and the overlapping regions between them are represented as edges.

**Untangle wrong overlaps with SNPs.** Once a long read OG is established, further optimization is necessary to obtain a strain-resolved OG,. In particular, the OG established contains a considerable amount of mistaken overlaps. The reasons for such mistaken overlaps to show are the high similarity between different strains, which leads to alignment algorithms aligning regions that look similar, but stem from different stains with each other. Therefore, assembling genomes using the still raw OG is prone to accumulating misassemblies and losses of strain-specific variations.

To enable HyLight to assemble strain-resolved contigs, it is therefore crucial to untangle such mistaken overlaps (i.e., removing the corresponding edges in the OG). To discover and untangle such overlaps, we utilize SNP information as per the steps displayed in Fig. 2.

First, for each read, we keep track of the mismatches and indels in comparison with the reads that aligned with it. Corresponding details can be immediately obtained from examining the CIGAR strings output by Minimap2. In other words, we leverage the mismatch information in combination with counts of reads that support the mismatch characters (nucleotides) to determine SNP sites.

There can be two cases: 1) A base in a read is different from the consensus base in all other reads, see 'T' at P1:X and 'C' at P2:X in Fig. 2. If less than 3 reads support such a base, we consider it a sequencing error. We do not break overlaps between reads due to this case. 2) There are two bases showing, and a sufficient amount of reads (≥3) that support each of them. See Fig. 2: this applies for both "A" and "C" at P1:X, "T" and "A" at P2:X, and "C" and "T" at P3:X. In any such case, we break overlaps that are affected by this scenario, in other words, we remove the corresponding edges from the OG.

In case of both 1) and 2) applying for pairs of aligned reads (for example for the read that contains "T" at P1:X in Fig. 2), we determine the identity of a read based on the SNP information referring to other positions, where bases of a read were not evaluated as sequencing errors (here P2:X and P3:X).

Note that Fig. 2 only refers to the case of reads from two different strains alone. In reality, scenarios encompass more than just two strains, however. Nevertheless, the rules that we have described can be straightforwardly extended to scenarios reflecting the presence of more than 2 strains: we distinguish between sequencing errors (supported by at most 2 reads) and strain-specific variation (supported by at least 3 reads) in the exact same way.

The encompassing evaluation of the SNP information obtained from studying the alignments of the overlapping part of reads eventually leads to unambiguous resolution of strain-aware overlap information, which leads to establishing a strain-aware OG as a result.

**Assemble long reads by strain aware OG and correct contigs.** After obtaining the strain-resolved graph, we can use Miniasm[53] to assemble contigs. Note that standard usage of Miniasm consists in providing an OG as immediately computed by Minimap2. The particular advance applied here is to break overlaps that connect reads from different strains before providing the OG as input to Miniasm. Despite prior distinguishing between errors and true variation during mistaken overlap removal, long reads often still contain numerous sequencing errors, which necessitates further correction of the contigs assembled by Miniasm using the strain-resolved OG we provided as input.

To do that, we align the long reads with the contigs and, subsequently, following the strategy described earlier, resolve erroneous overlaps between reads and contigs. After untangling the erroneous overlaps, we apply Racon[54] to correct the contigs further. As per its protocol, Racon cuts the mapped reads into 500bp windows and rapidly corrects contigs by way of a de Bruijn Graph (DBG) based procedure. It is important to note that both Miniasm and Racon originally lacked the ability to perform strain-aware assembly and correction. If the original, raw OG had been provided as input, both Miniasm and Racon would lose strain-specific variations during correction (Racon) and assembly (Miniasm). The outcome would be merged contigs affected by considerably more misassemblies. Since we provide a strain-resolved OG that has retained strain-specific variants correctly, HyLight is not affected by this issue.

**Module 2: Strain-aware assembly of short reads**
**Pick up and strain-aware assembly of unmapped short reads.** The low depth of long reads implies that still some sequencing errors have remained undetected. The low depth induces further that certain genomic regions or (even entire) strains cannot be reconstructed using long reads alone. To this end, we use short reads that remained unaligned with any of the long reads, to polish long read based contigs further, and fill gaps between the contigs. The specific steps are shown in Fig. 3. First, we use Minimap2 to align the short reads to the contigs generated from long reads, creating an OG. Then, based on the strategy mentioned above, we use SNPs to untangle the wrongly linked overlaps between two strains and obtain a strain-resolved OG. Reads that are not in this OG belong to regions or strains that have not yet been reconstructed. We then use the previously published method StrainXpress[12] to cluster these remaining short reads and perform strain aware assembly using the OG's emerging from the clustered short reads.

**Module 3: global assembly.** After having computed long-read and short-read contigs, we aim to extend them further by constructing a global contig graph through contig-to-contig alignment. Each vertex corresponds to a contig, and edges correspond to high-quality overlaps: overlaps are supposed to be longer than 100 bp showing at least 0.99 similarity, following guidance provided by previous works[37,67]). To reduce complexity, all transitive edges are removed, and contigs are joined into "branch-less" sequences. To identify branches, we evaluate short reads that match the corresponding contigs. After removing branches, HyLight updates the graph and extends contigs further. This process is iterated until no further branches are observed, resulting in the "master contigs" that establish HyLight's final output, available for downstream functional analysis.

**Synthetic data sets**
To compare the performance of different approaches, we utilized CAMISIM[56] (version 0.0.6) to produce four simulated hybrid

sequencing datasets consisting of Illumina MiSeq and PacBio CLR. These datasets included 3 *Salmonella* strains, 20 bacterial strains (10 species), 100 bacterial strains (30 species), and 210 bacterial strains (100 species), respectively. CAMISIM is a widely used metagenome simulator capable of modeling second and third-generation sequencing data with varying abundances and multi-sample time series based on real strain-level diversity.

The length of the simulated Illumina MiSeq reads is 2X250 bp, at an insert size of 450 bp. The N50 of PacBio CLR reads is 10 kbp at an average sequencing error rate of 10%. As per the principles of CAMISIM, the abundance of different strains is uneven, as sampled from a log-normal distribution. The average coverage of both Illumina MiSeq and PacBio CLR data for the four simulated communities is 20X and 10X respectively. The genomes of the 3 *Salmonella* strains were obtained from earlier work[68]. The genomes for the 20 bacterial strains, the 100 bacterial strains and the 210 bacterial strains communities were downloaded from an earlier study[55]. For details with respect to Genome ID's and their average nucleotide identity (ANI), please see Supplementary Data.

Additionally, to assess the impact of long read coverage, we generated six sequencing datasets by combining simulated and real data, which represents a typical simulation scenario known as "spike-in" data. This approach allows us to evaluate how methods assemble the simulated ("spiked-in") data, for which the ground truth is known, within a realistic context (although ground truth is lacking for the real data, hence the need to incorporate simulated reads). Specifically, we incorporated simulated reads from ten well-known *Salmonella* strains downloaded from[68] into six distinct real gut metagenome sequencing datasets. These datasets were obtained from experiments aiming at identifying functional characteristics of low-abundance and uncultured species in the human gut[69] (project number: PRJNA602101).

To simulate reads from the *Salmonella* strains, we utilized the CAMISIM simulator while closely matching the properties of the real sequencing data, ensuring optimal comparability. The synthesized short reads were set to a length of 2X150 bp. To account for the influence of read coverage, the coverage of synthesized long reads for the spiked-in strains varied from 5X to 30X in increments of 5X across the six real data sets. Each of the six spiked-in real data sets represented a specific coverage level for short reads. On the other hand, the coverage of the simulated NGS reads remained fixed at 20X across the six real data sets.

Considering the substantial number of reads, we randomly extracted 8,166,722 NGS reads and 181,092 TGS reads from each real hybrid sequencing data set for a less computationally intensive evaluation. These extracted reads were further processed for analysis. For more details regarding the ten *Salmonella* Genome ID's and SRA identifiers, please refer to Supplementary Data, "spike-in *Salmonella*".

**Real data sets**
We considered two microbial communities for which both TGS and NGS data were available in our experiments:

**Bmock12** is a mock community comprising 12 bacterial strains from 10 different species[70]. The mock community was sequenced using all ONT MinION, PacBio and Illumina sequencing platforms. The corresponding data sets were obtained from SRA (illumina SRR8073716, ONT SRR8351023, PacBio SRR8073714). The N50 read length for ONT and PacBio reads is 22,772 and 8,701, respectively. The Illumina reads have a read length of 2X150 bp, with an average insert size of 302.7 bp. It is worth noting that the number of reads mapped to *Micromonospora coxensis*, one of the 12 strains, was negligible[70], so we effectively dealt with only 11 bacterial strains. The average coverage for these 11 strains ranges from 74.56X to 3,093.79X, with a median of 1,376.35X. For the sake of a less runtime-intense evaluation in the light of the large amount of duplicates among the reads, we randomly extracted 20% of the reads, and further processed only these. Lastly, it

is important to address the challenges posed by this data set, which involve assembling the long reads of two species whose strains exhibit high average nucleotide identities (ANI). Specifically, this applies to the *Marinobacter* species and the *Halomonas* species, as they contain pairs of strains characterized by ANI values of 85% and 99%, respectively.

**NWCs.** The second real microbial community selected for analysis originates from natural whey starter cultures (NWCs)[71]. The metagenome samples of the NWCs were sequenced using Illumina MiSeq, generating reads with a length of 2 × 300 bp. Additionally, PacBio and ONT sequencing platforms were employed. We acquired the sequence data sets from SRA (illumina SRR7585899 and SRR7589561, ONT SRR7585900, PacBio SRR7589560). The N50 read lengths for PacBio and ONT TGS data are quite similar, measuring 11,895 and 9,562, respectively. In a previous study, complete genomes of six bacterial strains from three species were obtained[71]. The GenBank accession numbers for these six genomes are CP029252.1, CP031021.1, CP031024.1, CP031025.1, CP029252.1, and CP031021.1. We utilized these reference genomes as the ground truth for evaluating the accuracy of assembly results for distinct approaches.

In the NWCs datasets, we performed a removal of low-quality bases (> Q20), the NGS (Illumina) reads retained an unusually high error rate (indel error rate: 19.58/100 kbp; mismatch error rate: 883.28/100 kbp.). This does not reflect standard scenarios, and can have a considerable impact on the accuracy of the assembly. To restore a standard scenario, we corrected the NGS reads prior to hybrid assembly, by using bfc[72], an approved error corrector applicable for Illumina short reads.

**Three yeast strains.** To compare HyLight with Hifiasm-meta and metaMDBG, which are assemblers designed for high-quality PacBio HiFi reads, we mixed real reads of three yeast. Original datasets were intended for evaluating different assembly methods and sequencing platforms[59]. Consequently, the three yeasts Saccharomyces cerevisiae strain S288C, S. cerevisiae CICC-1445, and S. pombe FLO-DUT were simultaneously sequenced using PacBio HiFi, Oxford Nanopore Technologies, and the short-read sequencing technology BGISEQ (2 ×! 150 bp paired reads). Among these three datasets, S288C has an available haploid reference genome, allowing us to directly evaluate the assemblies using metaQUAST. For the other two yeasts, since no haploid reference genomes are available, we employed Merqury to assess the assembly results. The sequencing data for all three datasets are deposited in the SRA (S288C: PRJNA792930; CICC 1445: PRJNA792931; S. pombe FLO-DUT: PRJNA792932).

## QUAST evaluation criteria

During the evaluation process, we took into account all pertinent categories provided by MetaQUAST V5.1.0rc1[62], a widely recognized tool for assessing assembly quality. Following established guidelines, we incorporated the flags –ambiguity-usage all and –ambiguity-score 0.9999 specifically for the evaluation of metagenomic data. The remaining parameters were kept at their default values. In the subsequent sections, we will provide concise definitions of the metrics under consideration. For more comprehensive explanations, please refer to http://quast.sourceforge.net/docs/manual.html.

`Indels/100 kbp`. The sequence derived from raw PacBio CLR and ONT reads is susceptible to a significant presence of indel errors. In this context, "indels per 100 kbp" refers to the average count of insertion or deletion errors per 100,000 aligned bases within the contigs.

`Mismatches/100 kbp`. This metric represents the average count of mismatch errors per 100,000 aligned bases in the examined contigs.

`N/100 kbp`. This indicates the average count of uncalled bases (N's) per 100,000 bases in the evaluated contigs.

`Genome Fraction (GF)`. GF represents the proportion of aligned bases in the reference genome to which the contigs are aligned. Essentially, GF indicates the extent to which the evaluated contigs cover each of the strain-specific genomes.

`Misassembled contigs (MC)`. A contig is categorized as a *misassembled contig* if it contains one or more misassembly events. A misassembly event is identified when a contig aligns to the correct sequence but exhibits a gap larger than 1 kbp, an overlap exceeding 1 kbp with a different strand, or even with a distinct strain. The percentage of misassembled contig, relative to the total number of evaluated contig, is reported as "misassembled reads."

`NGA50`. NGA50 is defined as the longest contig length, where all contig of that length or longer align to at least 50% of the true sequence. In other words, NGA50 represents the maximum contig length that provides coverage to at least half of the true sequence through their alignments.

## Merqury evaluation criteria

Merqury is a reference-free approach for evaluating assembly results. It assesses the completeness and error rate of an assembly by analyzing the reproducibility of k-mers generated from the read data compared to those present in the assembled contigs.

`Completeness`. Completeness quantifies the fraction of reliable k-mers from the read data that are accurately represented in the assembled contigs. This metric reflects how comprehensively the assembled genome captures the full genetic content of the original DNA sample.

`Quality Value (QV)`. QV is a metric for evaluating the quality of assembled contigs, reflecting the confidence in their accuracy. A higher value indicates greater confidence in correctly identifying the contigs from the k-mers generated by NGS data, corresponding to better assembly results.

`Error rate`. In the context of Merqury, the reported error rate encompasses the cumulative frequencies of both indel error rate events and nucleotide mismatche error rate.

## Reporting summary

Further information on research design is available in the Nature Portfolio Reporting Summary linked to this article.

## Data availability

The simulated data generated in this study are available via Code Ocean, at https://doi.org/10.24433/CO.8025931.v2 and Zenodo, at https://doi.org/10.5281/zenodo.13295035. The real data used in this study have been deposited in the SRA database under accession codes SRR8073716 https://www.ncbi.nlm.nih.gov/sra/?term=SRR8073716 (Illumina), SRR8351023 https://www.ncbi.nlm.nih.gov/sra/?term=SRR8351023 (ONT), and SRR8073714 https://www.ncbi.nlm.nih.gov/sra/?term=SRR8073714 (PacBio) for Bmock 12, as well as accession codes SRR7585899 https://www.ncbi.nlm.nih.gov/sra/?term=SRR7585899 and SRR7589561 https://www.ncbi.nlm.nih.gov/sra/?term=SRR7589561 (Illumina), SRR7585900 https://www.ncbi.nlm.nih.gov/sra/?term=SRR7585900 (ONT) and SRR7589560 https://www.ncbi.nlm.nih.gov/sra/?term=SRR7589560 (PacBio) for NWCs. All other data supporting the findings described in this manuscript are available in the article and its Supplementary Information files.

## Code availability

The source code of HyLight is GPL-3.0 licensed, and can be retrieved at https://github.com/LuoGroup2023/HyLight. You also can reproduce the results or run your own data on Code Ocean, at https://doi.org/10.24433/CO.8025931.v2.

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

## Acknowledgements

AS received funding from the European Union's Horizon 2020 research and innovation program under Marie Skłodowska-Curie grant agreements No 956229 (ALPACA) and No 872539 (PANGAIA). XL was supported by the Natural Science Foundation of Hunan Province (Grant No. 2024JJ4008) and Fundamental Research Funds for the Central Universities (Grant No. 541109030062).

## Author contributions

X.K. and A.S. developed the method. X.K., X.L. and A.S. wrote the manuscript. X.K., W.Z., Y.L. and X.L. conducted the data analysis. X.K. implemented the software. All authors read and approved the final version of the manuscript.

## Funding

## Competing interests

The authors declare that they have no competing interests.
