## [Peer Review File · Nature Communications]

HyLight: Strain aware assembly of low coverage metagenomesREVIEWER COMMENTS

Reviewer #1 (Remarks to the Author):

This paper proposed a new strain aware assembly algorithm called HyLight, which improves the metagenome assembly by co-assembling both long noisy and short reads. Compared with existing metagenome assembly algorithms, HyLight presented better results on several real and simulated datasets.

Although the authors present decent results and examples for the community, I have several concerns and suggestions that might be addressed or discussed before this manuscript is suitable for publication.

Major specific comments:

1. The authors argued that HyLight is the only hybrid genome assembly algorithm that could make full use of the power of both types of reads with a cross-hybrid strategy, while all other tools are either short-read-first or long-read-first. However, the main difference of HyLight is the first step, where HyLight additionally takes short reads to correct long noisy reads. However, this additional step is very noisy, and cannot be easy to scale to large datasets. This is why most hybrid genome assembly algorithms do not incorporate this step. It would be better not to mention that this is a novel solution that could combine the power of both short and long noisy reads.
2. The authors said, "...In summary, sole application of TGS in metagenomics requires to substantially raise expenses: either to increase coverage (as for PacBio CLR and ONT) or because of employing more sophisticated sequencing protocols (PacBio HiFi or ONT reads of latest generations). In addition, these more sophisticated protocols entail substantial additional requirements in terms of computational resources..." It is hard to understand why assembling with PacBio HiFi reads is time-consuming. Currently, most new genome assembly projects are using PacBio HiFi reads since its assembly procedure is extremely fast. If I understand correctly, the authors argued that the consensus step of HiFi reads requires substantial computational resources. However, the PacBio company has already significantly accelerated it with GPU internally. It would be better not to mention this as the disadvantage of PacBio HiFi reads.
3. The authors said, "...Importantly, our protocols are not very affordable, but also lead to assemblies whose quality substantially pushes the limits of metagenome assembly from a global perspective..." If the cost does not matter, the authors should compare HyLight with new HiFi-based metagenome assembly algorithms like MetaMDBG and hifiasm-meta. Although they are not hybrid assembly algorithms, they should be able to natively produce strain aware results.
4. The authors said, "...From another perspective, our approach is also novel insofar as it is the only approach that makes use of overlap graphs, instead of DBG's..." However, hifiasm-meta is another metagenome assembly algorithm based on the overlap graph.
5. The authors said, "...Also, still—see the first reason—the quality of hybrid assemblies exceeds the quality of assemblies computed by such high-convenience protocols. The reason for the latter is the fact that TGS reads of the primary type are just still substantially (by factors of up to 2-4 times) longer." It is hard to understand since for ordinary single-sample genome assembly, the new HiFi reads could produce significantly better results. Although they are several times shorter than old long noisy reads, HiFi reads are much more accurate. If the authors think long noisy reads + short reads with HyLight could lead to better metagenome assemblies, they should add more experiments to demonstrate that.

Reviewer #1 (Remarks on code availability):

The code of HyLight is easy to be used.

Reviewer #2 (Remarks to the Author):

(See attached)

HyLight strain aware assembly

The authors present a hybrid/co-assembly method that leverages short and long read data in order to accurately recover genomes from a metagenome in a strain-aware fashion. The ideas contained in the manuscript are novel, especially the replacement of a de Bruijn graph approach for an overlap graph one, and the authors describe this “secrete sauce” approach quite nicely. This is in contrast to some assembly methods that don’t highlight the heuristics that lead to the largest amount of improvement in their assemblies.

The experiments included convincingly show that the HyLight approach significantly outperforms existing methods (be they strain-aware or not) in essentially all metrics. The real-world experiments still showed that HyLight outperforms existing methods in terms of genome fraction, but the overwhelming improvements seen in the simulated data did not completely translate to the real data, likely due to the fact, which the authors point out, that there is not much strain variation in the Bmock12 dataset. This is understandable though, since it is one of few (only?) real-world data sets that have both short and long read data with a known ground truth.

So while the scientific contribution of this paper is quite substantial, and I foresee this approach becoming a standard for metagenomic assembly in the future, the manuscript itself and the provided code serve to hamper the impact of this work. The manuscript itself is written in a way that significantly increases the cognitive load on the reader. There is a plethora of repetition (eg. 24 uses of “on the one hand/on the other hand”, missing text (eg. “?” when there should be a percentage), mislabeled sections (eg. the “Misassembled Contig Rate” section), and unfocused text. Specific comments illustrating these points are contained below.

Similarly, the software provided appears not to be in a workable state. There are two associated GitHub repositories, both containing slightly different code and installation instructions, neither of which worked for me. See the software review below for more details.

Given the resounding superiority of the HyLight approach in improving strain-aware hybrid assembly (or even just hybrid assembly itself), it would be disappointing if the exposition in the manuscript and the difficult-to-use code base lessened the impact of this approach.

General science comments/Questions:

1. The authors appear to be aware of DESMAN, as they used it to get strain compositions in their synthetic data. But nowhere is there a (brief) discussion of how DESMAN, as a tool to extract strains from metagenomes, compares with and contrasts to HyLight.

2. Related to the above, since CAMISIM was used in the generation of synthetic data, why do you need an external tool (DESMAN) to determine the strains, considering the ground truth produced by CAMISIM already supplies that?

General writing comments:

1. Lots of unnecessary language “on the one hand” “In contrast, however” etc. 24 uses of “on the one hand” + “on the other hand”
2. Overabundance of discourse markers/transitional phrases like “Rather,...” “Either,...” “In particular,...” “In a bit more detail, ...”, “Vice versa,...” All of which I observed in just a single paragraph-sized bit of text.
3. The introduction is quite verbose, sounding at times like a review paper. The detailed exposition on existing approaches seems better located outside of the Introduction (perhaps in a “Background” section). It is also repetitive: the authors briefly describe the approach towards the end of the introduction, then in more detail describe the workflow at the start of the “Results” section, then summarize it again in the first few (very short, but plentiful) paragraphs of the “Workflow” section, then in more detail in the rest of the “Workflow” section (line 263 onward), and then again in much more detail in the Methods; Workflow section.
4. Similar comments to the above can be said about the Discussion section.
5. There are dozens of single/double sentence paragraphs leading to fragmentation of ideas and negative impact on reader engagement.
6. Some figures/tables say “CrossHyLight” while the main manuscript calls it “HyLight”.
7. The section “Misassembled Contig Rate” doesn’t really discuss misassembled contig rate, and seems to interpret results before they are described “As a general trend...HyLight and StrainXpress considerably outperform Strainberry”.
8. I suggest avoiding recapitulating performance numbers and metrics in the text of the experiments section, particularly when those numbers exist in tables. Eg. Instead of the two very hard to read sentences:

“Both approaches are competitive in terms of strain awareness (Genome Fraction - HyLight: 90.16%; Strainberry: 78.56%; StrainXpress: ?%), where here, finally, HyLight also clearly outperforms StrainXpress. While Strainberry has drawbacks with respect to error rates (Indels - HyLight: 17.69/100kbp; Strainberry: 418.36/100kbp; StrainXpress: ?; Mismatches -HyLight: 52.78/100kbp; Strainberry: 102.55/100kbp; StrainXpress: ?), StrainXpress considerably trails in terms of contiguity (NGA50 -HyLight: 128015; Strainberry: 81842; StrainXpress: ?).”

Consider something like:

“As seen in <relevant tables or supplementary tables>, HyLight outperforms StrainXpress in terms of genome fraction, indels, mismatches, and contiguity. StrainXpress outperforms Strainberry in all these metrics save for contiguity where it trails considerably.” Or something in that style.

Specific writing comments:

1. Super chatty introduction: “we would like to present” -> “we present”
2. Typo: Page 3, line 129 “So. pursuing”
3. Typos: Page 3, line 130: backward quote (and in many other places)
4. Page 3: 138-140. Example of unnecessary verbosity. I suggest removing it.
5. Typo: Line 364: “that targets at all of”
6. Lines 360-> start of 364: more unnecessary repetition
7. Typo: Line 370: “To avoid repeating statements When”
8. Lots of “?” in the “Experiments” section where numbers should be (and one ç).
9. Inconsistent use of including the percentage sign or not (eg. line 386 vs 392-393)
10. I stopped keeping track of typos and the like around line 377 but the above should give an indication.

Software review:

The manuscript directs users to the repo: <https://github.com/HaploKit/HyLight> , however, the README of that repo indicates the user should clone the different repo <https://github.com/kangxiongbin/HyLight> which has different dependencies and is 9 commits ahead of the other.

I could not get the software to install.

1. Following the first link README omits the required `racon`, which was easy to fix with conda.
2. Running the example command on the example data resulted in a bunch of errors of the sort:

```
...
You need to rerun polyte in /scratch/dmk333_new/HyLight/example/tmp//fq_15000/10330/ or
decrease the size of cluster and rerun the whole steps
You need to rerun polyte in /scratch/dmk333_new/HyLight/example/tmp//fq_15000/10528/ or
decrease the size of cluster and rerun the whole steps
...
```
3. This suggested to me that it might be an issue with HaploConduct since the installation says that this is a dependency, but it is not included in the install directions of:

```
...
conda create -n HyLight
conda activate HyLight
conda install -c bioconda python=3.6 scipy pandas minimap2 bfc fmlrc2 ropebwt2 miniasm racon
...
```

I then noticed that:

 - a) HaploConduct cannot be installed in the same environment as the above one creates as it requires python version < 2.7.X while the install directions specify python=3.6.
 - b) There is some HaploConduct code in the HyLight repo (HyLight/tools/HaploConduct) and a makefile exists but no instructions in the readme. Even after making sure boost is installed (via `conda install conda-forge::boost`), running `make` results in other compilation errors like:

```
...
In file included from src/BranchReduction.h:25, from src/BranchReduction.cpp:13:
src/OverlapGraph.h:17:10: fatal error: boost/dynamic_bitset.hpp: No such file or directory
17 | #include <boost/dynamic_bitset.hpp>
```

```
| ~~~~~  
compilation terminated.  
src/EdgeCalculator.cpp:19:10: fatal error: boost/algorithm/string.hpp: No such file or  
directory  
19 | #include <boost/algorithm/string.hpp>  
| ~~~~~  
compilation terminated.  
...
```

There is also no description whatsoever about the output format or the expected output from the example. The authors also do not indicate anywhere how their results (eg. on the simulated and real data) can be reproduced.

In short, this repo appears to be a work in progress lacking many features:

1. Documentation of the code
2. Installation instructions that work (including the proper installation of dependencies)
3. Actual releases or deployment via a package manager such as conda
4. Tests to ensure proper installation and to increase confidence in the code base, etc.

I imagine in its current state, it would be difficult for this tool to find a wide audience of users.

We thank all Reviewers for their helpful comments. We are convinced that your input has considerably improved the manuscript. In the following, please find our detailed responses to your comments and questions.

Reviewer #1

1. The authors argued that HyLight is the only hybrid genome assembly algorithm that could make full use of the power of both types of reads with a cross-hybrid strategy, while all other tools are either short-read-first or long-read-first. However, the main difference of HyLight is the first step, where HyLight additionally takes short reads to correct long noisy reads. However, this additional step is very noisy, and cannot be easy to scale to large datasets. This is why most hybrid genome assembly algorithms do not incorporate this step. It would be better not to mention that this is a novel solution that could combine the power of both short and long noisy reads.

Thanks for pointing this out. We are not sure, however, whether there is a bit of a misunderstanding. We do not only use short reads to correct errors in long reads, and then assemble the corrected long reads. This would reflect a well-known strategy (one subtype of long-read first).

Here, in addition, we also assemble short reads, and use them to complement the (corrected) long-read based assemblies. For example, the short read assemblies can fill gaps in the long read assemblies, or resolve disambiguities of other nature that affect the long read assemblies.

So, we respectfully insist on pursuing a “novel” strategy. In comparison to short-read first and long-read first, it is both of that in combination. Such a combination of the two strategies had not yet been presented in the literature before.

In the revised version of the manuscript, we try to put particular emphasis on what is novel about our strategy.

2. The authors said, “...In summary, sole application of TGS in metagenomics requires to substantially raise expenses: either to increase coverage (as for PacBio CLR and ONT) or because of employing more sophisticated sequencing protocols (PacBio HiFi or ONT reads of latest generations). In addition, these more sophisticated protocols entail substantial additional requirements in terms of computational resources...” It is hard to understand why assembling with PacBio HiFi reads is time-consuming. Currently, most new genome assembly projects are using PacBio HiFi reads since its assembly procedure is extremely fast. If I understand correctly, the authors argued that the consensus step of HiFi reads requires substantial computational resources. However, the PacBio company has already significantly accelerated it with GPU internally. It would be better not to mention this as the disadvantage of PacBio HiFi reads.

Yes, thanks for the comment. We agree. You say “most new genome assembly projects are using PacBio HiFi reads since its assembly procedure is extremely fast”,

and we agree on that point as well. We would add that most new genome assembly projects use PacBio HiFi reads because of the convenience of the procedure. However, you can compare this to buying an easily steerable hi-speed race car: this comes at a high price, so is not affordable by lots of lesser well equipped sequencing laboratories worldwide (of which there are plenty!).

Obviously, running experiments with PacBio HiFi reads in a less effortless manner requires particular equipment (GPU's!). However, this now requires hardware-aware adaptations, which, again, many laboratories have not yet established in a sufficient way, in particular if they are not sufficiently budgeted (which, as we point out, prevents effortless execution of strain-level metagenome analyses in the majority of places worldwide). In other words, PacBio HiFi reads, however fast these are in the particular environment they require, are demanding in terms of computational resources. The fact that PacBio worked on exactly that point just testifies this very issue of PacBio HiFi. It is our opinion that, notwithstanding our deep respect for the technology itself, the level of sophistication and the hard- and software-hungry design that drives the technology will prevent PacBio HiFi to get to the level of ease in terms of underlying technology really soon. Again, that doesn't mean it's a bad idea to use it when you can afford it—on the contrary.

We keep arguing that it is a major advantage of our approach: it requires hardly anything in this respect. Ordinary CPU's do the job in sufficiently little time, and the sequencing itself draws considerably less monetary expenses as well.

3. The authors said, "...Importantly, our protocols are not very affordable, but also lead to assemblies whose quality substantially pushes the limits of metagenome assembly from a global perspective..." If the cost does not matter, the authors should compare HyLight with new HiFi-based metagenome assembly algorithms like MetaMDBG and hifiasm-meta. Although they are not hybrid assembly algorithms, they should be able to natively produce strain aware results.

Thanks again for pointing this out, and our apologies. The sentence above contains a (totally misleading) typo: an "only" is missing. It should read "... our protocols are not only very affordable, but also lead to assemblies ...". In other words, our protocols are considerably more affordable in terms of monetary expenses and requirements in terms of runtime / platform-specifics. This is one of the big arguments: nearly every sequencing laboratory, also in third-world countries, can afford to run our methods. (where we understand that driving a luxury hi-speed car is more fun—but what to do if you cannot afford buying such a car?)

Regardless of this possible misunderstanding, in order to further erase any related doubts, we nevertheless ran HyLight on benchmark datasets on which one could compare it with HiFiasm_meta and MetaMDBG. We thank the Reviewer another time, because we agree that such experiments were missing, at any rate.

So, we evaluated the performance of HyLight, HiFiasm-meta, and MetaMDBG using recently published benchmark datasets (<https://doi.org/10.1093/bib/bbac146>).

These included the genomes of three yeast strains sequenced with all NGS, ONT, and HiFi technologies. The S288C strain, as one of the three strains, provided a haplotype resolution reference genome, which enables one to evaluate the results using Quast. Since the other two genomes lack a reference sequence, evaluating results using Merqury was an appropriate choice.

Assessment of the assembled genomes of the three yeast strains revealed that HyLight exhibited higher completeness and lower error rates compared to HiFiasm-meta and MetaMDBG. We recall that while HyLight utilizes NGS and ONT data, HiFiasm-meta and MetaMDBG are designed to operate on high-accuracy HiFi reads alone. HyLight demonstrated superior performance across the relevant benchmark criteria (Details are presented in Table 4 and Supp. Table 9).

4. The authors said, "...From another perspective, our approach is also novel insofar as it is the only approach that makes use of overlap graphs, instead of DBG's..." However, hifiasm-meta is another metagenome assembly algorithm based on the overlap graph.

Thank you for pointing this out. We introduced a misunderstanding here. Note that also earlier metagenome assembly approaches of ours are overlap graph based (e.g. "StrainXpress"), so we are well aware of extant overlap graph based approaches.

We need to be more specific here. We wanted to make the claim that we present the first overlap graph based *hybrid* metagenome assembly based approach. We have corrected this in the manuscript accordingly.

5. The authors said, "...Also, still—see the first reason—the quality of hybrid assemblies exceeds the quality of assemblies computed by such high-convenience protocols. The reason for the latter is the fact that TGS reads of the primary type are just still substantially (by factors of up to 2-4 times) longer." It is hard to understand since for ordinary single-sample genome assembly, the new HiFi reads could produce significantly better results. Although they are several times shorter than old long noisy reads, HiFi reads are much more accurate. If the authors think long noisy reads + short reads with HyLight could lead to better metagenome assemblies, they should add more experiments to demonstrate that.

Thank you for this comment. We agree that we had not provided sufficient evidence. Please see the discussion of the experiments we added as the answer to your comment before. As you say, ONT reads of the older generation tend to be longer than HiFi reads. For reasons we do not understand, this was not necessarily the case for these benchmark datasets (note that metagenomes sequenced with all of TGS, NGS and HiFi are rare). Nevertheless, it appears that noisy long reads + short reads lead to assemblies that are better than those computed with HiFi reads alone. See Table 4 and Supp Table 9 for the corresponding results.

Reviewer #2

General science comments/Questions:

1. The authors appear to be aware of DESMAN, as they used it to get strain compositions in their synthetic data. But nowhere is there a (brief) discussion of how DESMAN, as a tool to extract strains from metagenomes, compares with and contrasts to HyLight.

Thank you for pointing this out: in our writing, we were not clear about the usage of DESMAN and the data provided through the corresponding publication. There are two things that relate to each other.

First, we did not use DESMAN to infer the composition of strains in our synthetic data. Rather, the information on strain content was provided by the authors of DESMAN: they selected strains such that the resulting mix of strains matched composition of strains found in natural environmental settings.

Second, DESMAN does not assemble genomes. Rather, DESMAN employs external assembly software such as Megahit or Spades (reflecting preferences in the DESMAN paper) to assemble the genomes in a first step. Subsequently, DESMAN makes use of the resulting SNP information to predict both the number of strains in the dataset and their abundances.

Because DESMAN itself does not assemble genomes, but depends on assembly software itself, a comparison with genuine assembly approaches would only have very limited meaning. Most likely, it would even create the misleading impression that DESMAN is an assembly approach, although it classifies as a post-processing tool.

2. Related to the above, since CAMISIM was used in the generation of synthetic data, why do you need an external tool (DESMAN) to determine the strains, considering the ground truth produced by CAMISIM already supplies that?

We believe that there is a misunderstanding. CAMISIM generates synthetic reads based on (here: strain-level) ground truth genomes provided as input. These ground truth genomes are the ones retrieved from the DESMAN publication (but not generated using DESMAN, see above).

We have applied major changes to the manuscript in various places to clarify what we are trying to achieve in our work.

General writing comments:

Thank you very much for your efforts with respect to improving the language of our manuscript. While we are certainly fluent in English, we are not native speakers. So, many of the flaws you mention just escape our attention when proofreading the manuscript. Unavoidably, one tends to mostly focus on the transport of scientific

concepts, but not necessarily on the flow of texts (which hardly ever happens in one's native language). Again, we really mean to thank you for this.

1. Lots of unnecessary language “on the one hand” “In contrast, however” etc. 24 uses of “on the one hand” + “on the other hand”

We have done our best to fix such issues.

2. Overabundance of discourse markers/transitional phrases like “Rather,...” “Either,...” “In particular,...” “In a bit more detail, ...”, “Vice versa,...” All of which I observed in just a single paragraph-sized bit of text.

We have done our best also in this respect.

3. The introduction is quite verbose, sounding at times like a review paper. The detailed exposition on existing approaches seems better located outside of the Introduction (perhaps in a “Background” section). It is also repetitive: the authors briefly describe the approach towards the end of the introduction, then in more detail describe the workflow at the start of the “Results” section, then summarize it again in the first few (very short, but plentiful) paragraphs of the “Workflow” section, then in more detail in the rest of the “Workflow” section (line 263 onward), and then again in much more detail in the Methods; Workflow section.

We have created a “Background” subsection and added that to the end of the Introduction, without that reading the Introduction requires knowledge of the corresponding contents.

As for repetitiveness with respect to descriptions of our workflow, there are no more descriptions in the Introduction and at the start of Results. As for descriptions in the Workflow section and in Methods, we decided to keep these two versions as they are. While the goal of the version in Results is to provide an overview of our protocol as a methodical achievement, the goal of the version in Methods is to guarantee full conceptual reproducibility. Some parts of the first version are re-stated in Methods to guarantee the contextual consistency of the respective texts there, which we find necessary for seamless understanding of the corresponding passages.

4. Similar comments to the above can be said about the Discussion section.

We have done our best to take care of the Discussion section in terms of the improvements that you had suggested in general.

5. There are dozens of single/double sentence paragraphs leading to fragmentation of ideas and negative impact on reader engagement.

We have done our best to “defragmentize” ideas by regrouping and merging paragraphs.

6. Some figures/tables say “CrossHyLight” while the main manuscript calls it “HyLight”.

We have changed everything to the intended “HyLight”.

7. The section “Misassembled Contig Rate” doesn’t really discuss misassembled contig rate, and seems to interpret results before they are described “As a general trend...HyLight and StrainXpress considerably outperform Strainberry”.

Thanks. The section was not clear; in fact, it did discuss misassembled contig rate. We have reformulated the respective sentences.

8. I suggest avoiding recapitulating performance numbers and metrics in the text of the experiments section, particularly when those numbers exist in tables. Eg. Instead of the two very hard to read sentences:

“Both approaches are competitive in terms of strain awareness (Genome Fraction - HyLight: 90.16%; Strainberry: 78.56%; StrainXpress: ?%), where here, finally, HyLight also clearly outperforms StrainXpress. While Strainberry has drawbacks with respect to error rates (Indels -HyLight: 17.69/100kbp; Strainberry: 418.36/100kbp; StrainXpress: ?; Mismatches -HyLight: 52.78/100kbp; Strainberry: 102.55/100kbp; StrainXpress: ?), StrainXpress considerably trails in terms of contiguity (NGA50 -HyLight: 128015; Strainberry: 81842; StrainXpress: ?).”

Consider something like:

“As seen in <relevant tables or supplementary tables>, HyLight outperforms StrainXpress in terms of genome fraction, indels, mismatches, and contiguity. StrainXpress outperforms Strainberry in all these metrics save for contiguity where it trails considerably.” Or something in that style.

We have applied considerable changes to the corresponding paragraphs to get rid of such hard-to-follow listings of numbers, as above-cited.

Specific writing comments:

1. Super chatty introduction: “we would like to present” -> “we present”
2. Typo: Page 3, line 129 “So. pursuing”
3. Typos: Page 3, line 130: backward quote (and in many other places)
4. Page 3: 138-140. Example of unnecessary verbosity. I suggest removing it.
5. Typo: Line 364: “that targets at all of”
6. Lines 360-> start of 364: more unnecessary repetition
7. Typo: Line 370: “To avoid repeating statements When”
8. Lots of “?” in the “Experiments” section where numbers should be (and one ¿).
9. Inconsistent use of including the percentage sign or not (eg. line 386 vs 392-393)
10. I stopped keeping track of typos and the like around line 377 but the above should give an indication.

Thank you one more time for your careful reading. We have done our best to take care of all issues mentioned.

Software review:

The manuscript directs users to the repo: <https://github.com/HaploKit/HyLight> , however, the README of that repo indicates the user should clone the different repo <https://github.com/kangxiongbin/HyLight> which has different dependencies and is 9 commits ahead of the other.

I could not get the software to install.

1. Following the first link README omits the required `racon`, which was easy to fix with conda.
2. Running the example command on the example data resulted in a bunch of errors of the sort:

...

You need to rerun polyte in

`/scratch/dmk333_new/HyLight/example/tmp//fq_15000/10330/` or decrease the size of cluster and rerun the whole steps

You need to rerun polyte in

`/scratch/dmk333_new/HyLight/example/tmp//fq_15000/10528/` or decrease the size of cluster and rerun the whole steps

...

3. This suggested to me that it might be an issue with HaploConduct since the installation says that this is a dependency, but it is not included in the install directions of:

...

```
conda create -n HyLight
```

```
conda activate HyLight
```

```
conda install -c bioconda python=3.6 scipy pandas minimap2 bfc fmlrc2 ropebwt2 miniasm racon
```

...

I then noticed that:

a) HaploConduct cannot be installed in the same environment as the above one creates as it requires python version $< 2.7.X$ while the install directions specify python=3.6.

b) There is some HaploConduct code in the HyLight repo (HyLight/tools/HaploConduct) and a makefile exists but no instructions in the readme. Even after making sure boost is installed (via `conda install conda-forge::boost`), running `make` results in other compilation errors like:

...

```
In file included from src/BranchReduction.h:25, from src/BranchReduction.cpp:13:
src/OverlapGraph.h:17:10: fatal error: boost/dynamic_bitset.hpp: No such file or
directory
```

```
17 | #include
```

```
| ^~~~~~
```

```
compilation terminated.
```

```
src/EdgeCalculator.cpp:19:10: fatal error: boost/algorithm/string.hpp: No such file or
directory
19 | #include
   | ^~~~~~
compilation terminated.
...

```

There is also no description whatsoever about the output format or the expected output from the example. The authors also do not indicate anywhere how their results (eg. on the simulated and real data) can be reproduced.

In short, this repo appears to be a work in progress lacking many features:

1. Documentation of the code
2. Installation instructions that work (including the proper installation of dependencies)
3. Actual releases or deployment via a package manager such as conda
4. Tests to ensure proper installation and to increase confidence in the code base, etc.

I imagine in its current state, it would be difficult for this tool to find a wide audience of users.

P.S. The DOCX version is attached to this review for ease of use in revisions.

Thank you very much for installing and testing our HyLight.

We have made corresponding modifications on GitHub based on your suggestions, introducing the output file types and indicating which one is the final output file.

We have addressed the issues you encountered and documented how to resolve similar issues on GitHub.

We have now unified the link: <https://github.com/LuoGroup2023/HyLight.git>

Additionally, we also provide a Docker link on GitHub to facilitate other users in directly installing and setting up the HyLight environment.

REVIEWERS' COMMENTS

Reviewer #1 (Remarks to the Author):

The authors argued that current methods utilizing PacBio HiFi reads require specific high-end GPUs, which may not be affordable for many users. However, it should be noted that ONT reads, the inputs for HyLight, also require GPU support for base-calling, which is a time-consuming process as well. Given this, I suggest that the discussion regarding the affordability and hardware requirements be reconsidered or removed from the manuscript, as it presents an unfair comparison. All my other comments have been addressed.

Reviewer #1 (Remarks on code availability):

HyLight could be run with the datasets used in the manuscript by Docker.

Reviewer #3 (Remarks to the Author):

The authors have developed a hybrid strain-aware metagenome assembly approach named HyLight that integrates both long-read and short-read data. I appreciate that the authors have addressed most of the previous comments. However, I still have the following comments and questions:

General Science Comments:

1. Line 19: The term "qne" appears to be a typo. Did you mean "one"?
2. Lines 60-61: I wouldn't say PacBio (HiFi) or ONT (Q30+) are "error-free". A more accurate description would be "have very low error rates".
3. Although a Code Ocean capsule for DeChat was provided in the review, the manuscript does not mention DeChat. The methods section only refers to the use of FMLRC2 for error correction of long reads. Could you please clarify this discrepancy?
4. I couldn't find any links to download the synthetic datasets mentioned in the manuscript. I suggest uploading these datasets to a public repository like Zenodo, allowing users to easily access and reproduce the results.

General Writing Comments:

1. Some parts of the manuscript contain overly informal language. I recommend to avoid such language in scientific writing. Some places I noticed are,
 - Line 24: "TGS data of the cheap kind" could be rephrased.
 - Line 78: The phrase "of whatever kind" is too casual.
 - Line 784 "a.k.a." could be replaced with "i. e."
2. Starting paragraphs with the word "this" can sometimes be unclear on what it is referred to. Consider combining these sentences with the preceding paragraph for better logical flow.
3. Some sentences are too long spanning over 3-4 lines and are difficult to follow. It would be better to break them into shorter, more concise sentences. For example, in lines 88-91: "While StrainXpress assemblies remain too fragmented, Strainberry requires elevated coverage rates for the TGS reads (which, as repeatedly pointed out, is expensive), which confirms that application of only NGS or

application of only TGS is insufficient from also a methodological point of view." This could be split into two sentences for clarity.

4. In the section Results  Workflow  Second module, there is an inconsistency in pronoun usage. The steps initially use "we" (we align, we inspect, etc.), but from point 4, they switch to third person (one is, one concludes, one collects, one discards, etc.). I suggest using "we" or referring to the tool name "HyLight."

5. Some references are repeated in the text. For instance, in line 329, reference 12 appears multiple times.

6. Please double-check that the values mentioned in the text match those in the tables. For example, in line 468, the genome fraction of HyLight is noted as 90.95%, while Table 1 lists it as 90.49%.

7. All genus/species names should be italicised. For example, the yeast species names in lines 641-642 are not italicised.

Specific Writing Comments:

1. Line 48: Change "small error rates" to "low error rates"
2. Line 72: "on the one hand" should be "on one hand"
3. Line 102: Replace "pieces of sequence" with "sequences" or "fragments"
4. Line 129: "are the by far predominant assembly paradigm" should be "are by far the predominant assembly paradigm"
5. Line 168: "limits of the possible in" should be "limits of the possibility in"
6. Line 273: "genuinely strain-specific sequence" should be "genuine strain-specific sequences"
7. Line 300: "Intro" should be "Introduction"
8. Line 353: "miassembled" should be "misassembled"
9. Line 362: "target at the accuracy" should be "target the accuracy"
10. Line 398: Remove the inverted question mark ¿.
11. Line 693: "texonomic" should be "taxonomic"
12. Line 714: "earlier type of" should be "earlier types of"
13. Lines 736-737: Correct the double backwards quotes in "short-read-first" and "long-read-first"
14. Lines 791-794: Correct the backwards quotes in 'T', 'C' and 'A' (and in many other places).

I highly recommend the authors thoroughly proofread the manuscript and address these issues to enhance the clarity and professionalism of the paper.

Reviewer #3 (Remarks on code availability):

Software Review Comments:

I appreciate the improvements made to the GitHub repository and I was able to install and run HyLight on the example data provided. However, the instructions provided on the GitHub repository could be further improved in terms of clarity, organisation and software best practices.

1. The phrase "HyLight relies on the following dependencies:" appears twice.
2. Information about input and output formats is provided in the "Examples" section. Please consider creating separate sections such as "Inputs," "Running HyLight," and "Outputs" that make logical sense when running the tool.
3. The command "sh install.sh" is executed within the "HyLight" folder, but the subsequent example

command runs HyLight from `./script/HyLight.py`. There is no instruction to change to the "example" directory before running this command, which may be confusing for novice users.

4. Currently, the instructions are using relative paths to execute `HyLight.py`, requiring users to type the full path to `HyLight.py`. It would be more user-friendly to package and install the code in the system path, allowing execution just using `HyLight.py` from any directory. A popular technique is to use `setuptools` and create a `setup.py` to install the tool using `pip install`. Authors can refer to Python packaging resources such as <https://packaging.python.org/en/latest/tutorials/packaging-projects/>. Once set up, it should be straightforward to publish to repositories such as Bioconda and PyPI, which will greatly expand the tool's reach.

5. It would be helpful to include resources or instructions on converting R1/R2 reads into interleaved FASTQ format.

6. An explanation of the various parameters for HyLight should be included, as they may not be intuitive for all users. Adding the command `HyLight.py --help` to the README to list all parameters with explanations would be beneficial.

7. The README should explain all output files, such as `short_stageb.fasta` and `all_contigs.fa`, aligning with comment #2 about organising the README.

8. This is an additional suggestion to improve the documentation of HyLight. Please consider creating detailed documentation hosted on a platform like a Wiki, GitHub Pages, or Read the Docs. This documentation can include sections on installation, inputs, outputs, running the tool, frequently asked questions, etc., making it easier for users to navigate and understand the tool.

Response to Reviewers

We would like to thank all referees for the additional comments. We have addressed all remaining concerns.

Reviewer #1 (Remarks to the Author)

The authors argued that current methods utilizing PacBio HiFi reads require specific high-end GPUs, which may not be affordable for many users. However, it should be noted that ONT reads, the inputs for HyLight, also require GPU support for base-calling, which is a time-consuming process as well. Given this, I suggest that the discussion regarding the affordability and hardware requirements be reconsidered or removed from the manuscript, as it presents an unfair comparison. All my other comments have been addressed.

We acknowledge the point regarding the comparison of hardware requirements between PacBio HiFi reads and ONT reads. To address this concern, we have removed the discussion related to affordability and hardware requirements from the manuscript. We appreciate the thorough review and constructive comments, which have significantly improved our manuscript.

Reviewer #3 (Remarks to the Author)

The authors have developed a hybrid strain-aware metagenome assembly approach named HyLight that integrates both long-read and short-read data. I appreciate that the authors have addressed most of the previous comments. However, I still have the following comments and questions:

General Science Comments:

1. Line 19: The term "qne" appears to be a typo. Did you mean "one"?

Corrected

2. Lines 60-61: I wouldn't say PacBio (HiFi) or ONT (Q30+) are "error-free". A more accurate description would be "have very low error rates".

Done

3. Although a Code Ocean capsule for DeChat was provided in the review, the manuscript does not mention DeChat. The methods section only refers to the use of FMLRC2 for error correction of long reads. Could you please clarify this discrepancy?

We are not sure why DeChat was provided in the review, as our manuscript does not involve this method. DeChat is another project of our collaborator, Xiao Luo, whose method is currently under submission and has not yet been published. It is possible that the editor recommended you review that other submission as well.

4. I couldn't find any links to download the synthetic datasets mentioned in the manuscript. I suggest uploading these datasets to a public repository like Zenodo, allowing users to easily access and reproduce the results.

Thanks for your reminder. We upload synthetic datasets in Zenodo and Ocean code. The link of them is in the new version manuscript.

General Writing Comments:

1. Some parts of the manuscript contain overly informal language. I recommend to avoid such language in scientific writing. Some places I noticed are,

- Line 24: "TGS data of the cheap kind" could be rephrased.

Done

- Line 78: The phrase "of whatever kind" is too casual.

Done

- Line 784 "a.k.a." could be replaced with "i. e."

Done

2. Starting paragraphs with the word "this" can sometimes be unclear on what it is referred to. Consider combining these sentences with the preceding paragraph for better logical flow.

Done

3. Some sentences are too long spanning over 3-4 lines and are difficult to follow. It would be better to break them into shorter, more concise sentences. For example, in lines 88-91: "While StrainXpress assemblies remain too fragmented, Strainberry requires elevated coverage rates for the TGS reads (which, as repeatedly pointed out, is expensive), which confirms that application of only NGS or application of only TGS is insufficient from also a methodological point of view." This could be split into two sentences for clarity.

Done

4. In the section Results  Workflow  Second module, there is an inconsistency in pronoun usage. The steps initially use "we" (we align, we inspect, etc.), but from point 4, they switch to third person (one is, one concludes, one collects, one discards, etc.). I suggest using "we" or referring to the tool name "HyLight."

Done

5. Some references are repeated in the text. For instance, in line 329, reference 12 appears multiple times.

Done

6. Please double-check that the values mentioned in the text match those in the tables. For

example, in line 468, the genome fraction of HyLight is noted as 90.95%, while Table 1 lists it as 90.49%.

Corrected

7. All genus/species names should be italicised. For example, the yeast species names in lines 641-642 are not italicised.

Corrected

Specific Writing Comments:

1. Line 48: Change "small error rates" to "low error rates"

Done

2. Line 72: "on the one hand" should be "on one hand"

Done

3. Line 102: Replace "pieces of sequence" with "sequences" or "fragments"

Done

4. Line 129: "are the by far predominant assembly paradigm" should be "are by far the predominant assembly paradigm"

Done

5. Line 168: "limits of the possible in" should be "limits of the possibility in"

Done

6. Line 273: "genuinely strain-specific sequence" should be "genuine strain-specific sequences"

Done

7. Line 300: "Intro" should be "Introduction"

Done

8. Line 353: "miassembled" should be "misassembled"

Corrected

9. Line 362: "target at the accuracy" should be "target the accuracy"

Corrected

10. Line 398: Remove the inverted question mark.

Done

11. Line 693: "texonomic" should be "taxonomic"

Corrected

12. Line 714: "earlier type of" should be "earlier types of"

Corrected

13. Lines 736-737: Correct the double backwards quotes in "short-read-first" and "long-read-first"

Done

14. Lines 791-794: Correct the backwards quotes in 'T', 'C' and 'A' (and in many other places).

Corrected

Reviewer #3 (Remarks on code availability):

Software Review Comments:

I appreciate the improvements made to the GitHub repository and I was able to install and run HyLight on the example data provided. However, the instructions provided on the GitHub repository could be further improved in terms of clarity, organisation and software best practices.

1. The phrase "HyLight relies on the following dependencies:" appears twice.

We have removed the duplicate phrase "HyLight relies on the following dependencies:", ensuring it appears only once in the documentation.

2. Information about input and output formats is provided in the "Examples" section. Please consider creating separate sections such as "Inputs," "Running HyLight," and "Outputs" that make logical sense when running the tool.

We appreciate your suggestion for better organization. We have restructured the README to include separate sections for "Inputs", "Running HyLight", and "Outputs", providing a more logical flow of information for users.

3. The command "sh install.sh" is executed within the "HyLight" folder, but the subsequent example command runs HyLight from "../script/HyLight.py." There is no instruction to change to the "example" directory before running this command, which may be confusing for novice users.

We have added clear instructions to change to the "example" directory before running the HyLight command. This addition should help prevent confusion, especially for novice users.

4. Currently, the instructions are using relative paths to execute HyLight.py, requiring users to type the full path to HyLight.py. It would be more user-friendly to package and install the code in the system path, allowing execution just using "HyLight.py" from any directory. A popular technique is to use "setuptools" and create a setup.py to install the tool using "pip install". Authors can refer to Python packaging resources such as <https://packaging.python.org/en/latest/tutorials/packaging-projects/>. Once set up, it should be straightforward to publish to repositories such as Bioconda and PyPI, which will greatly expand the tool's reach.

Thank you for your valuable suggestion. We have taken your advice into consideration and have now uploaded HyLight to Bioconda. Users can now easily install HyLight with a single command: `conda install hylight -c bioconda`. We have also updated our GitHub repository to reflect this change and provide instructions for this new installation method.

5. It would be helpful to include resources or instructions on converting R1/R2 reads into interleaved FASTQ format.

We have added instructions on using fastp to merge R1/R2 reads into interleaved FASTQ format, including specific command examples.

6. An explanation of the various parameters for HyLight should be included, as they may not be intuitive for all users. Adding the command "HyLight.py --help" to the README to list all parameters with explanations would be beneficial.

We have included the command "HyLight.py --help" in the README, which lists all parameters with their explanations, making it easier for users to understand the available options.

7. The README should explain all output files, such as "short_stageb.fasta" and "all_contigs.fa," aligning with comment #2 about organising the README.

We have updated the README to include explanations for all output files, including "short_stageb.fasta" and "all_contigs.fa", as per your recommendation.

8. This is an additional suggestion to improve the documentation of HyLight. Please consider creating detailed documentation hosted on a platform like a Wiki, GitHub Pages, or Read the Docs. This documentation can include sections on installation, inputs, outputs, running the tool, frequently asked questions, etc., making it easier for users to navigate and understand the tool.

Thank you for your suggestion regarding HyLight's documentation.

While we understand the benefits of extensive documentation, we believe our current GitHub README sufficiently covers HyLight's needs. HyLight is designed to be straightforward, typically requiring only a few parameter adjustments. The existing documentation includes all necessary sections: installation, inputs, outputs, usage, and FAQs.

We aim to balance informativeness with simplicity. However, we remain open to user feedback and will consider expanding our documentation if user needs or tool complexity increase in the future.

Thank you for your interest in improving HyLight. We value your feedback and welcome any further suggestions.